# EXO1 promotes the meiotic MLH1-MLH3 endonuclease through conserved interactions with MLH1, MSH4 and DNA

Megha Roy [1,6], Aurore Sanchez [1,5,6] ✉, Raphael Guerois [2], Issam Senoussi[1,3], Arianna Cerana[4], Jacopo Sgrignani[1], Andrea Cavalli [1], Andrea Rinaldi[4] & Petr Cejka [1] ✉

The endonuclease activity of MLH1-MLH3 (MutLγ) is stimulated by MSH4-MSH5 (MutSγ), EXO1, and RFC-PCNA to resolve meiotic recombination intermediates such as double Holliday junctions (HJs) into crossovers. We show that EXO1 directly interacts with MLH1 via the EXO1 MIP motif and a patch centered around EXO1-I403. Disrupting this interaction unexpectedly only partially inhibited MutLγ. We found that EXO1 also directly interacts with MutSγ. Crucially, a single point mutation in EXO1 (W371E) impairs its interaction with MSH4 and completely abolished its ability to activate DNA nicking by MutLγ without affecting its intrinsic nuclease function. Finally, disrupting magnesium coordinating residues in the nuclease domain of EXO1 has no impact on MutSγ-MutLγ activity, while the integrity of EXO1 residues mediating interactions with double-stranded DNA (dsDNA) is important. Our findings suggest EXO1 is an integral structural component of the meiotic resolvase complex, supported by conserved interactions with MutSγ, MutLγ and dsDNA. We propose that EXO1 helps tether MutSγ-MutLγ to dsDNA downstream of HJ recognition to promote DNA cleavage.

Most eukaryotic organisms employ meiosis to produce haploid gametes[1]. During the prophase of the first meiotic division, the SPO11 complex induces DNA double-strand breaks (DSBs)[2]. The breaks are repaired by resection, followed by strand invasion into a matched DNA molecule[3], which in meiosis is often the homologous chromosome. Subsequently, the joint molecules mature into intermediates termed double Holliday junctions (dHJs). A central outcome of meiotic homologous recombination is the formation of crossovers, which represent reciprocal exchanges of DNA segments between the recombining chromosomes, and thus promote genetic diversity[1]. Recombination in most organisms results in a physical attachment (chiasmata) between homologous chromosomes, which is required for proper chromosome segregation during the first meiotic division.

It is believed that most meiotic crossovers result from biased resolution of dHJs. The majority of these crossovers depend on the nuclease activity of MLH1-MLH3 (MutLγ)[4–12]. MutLγ-dependent crossovers exhibit interference, i.e., are evenly distributed along the chromosomal DNA, and are referred to as Class I crossovers. In contrast, Class II crossovers do not exhibit interference, and are thought to result from the action of structure-selective nucleases, such as MUS81, which cleave dHJs in a random orientation, resulting in both crossover and non-crossover events[5,13,14]. The biased resolution of meiotic dHJs into crossovers by MutLγ requires a number of co-factors, including

[1]Institute for Research in Biomedicine, Università della Svizzera italiana (USI), Faculty of Biomedical Sciences, Bellinzona, Switzerland. [2]Université Paris-Saclay, CEA, CNRS, Institute for Integrative Biology of the Cell (I2BC), Gif-sur-Yvette, France. [3]Department of Biology, Institute of Biochemistry, Eidgenössische Technische Hochschule (ETH), Zürich, Switzerland. [4]Institute of Oncology Research, Università della Svizzera italiana (USI), Faculty of Biomedical Sciences, Bellinzona, Switzerland. [5]Present address: Institut Curie, PSL University, Sorbonne Université, CNRS UMR3244, Paris, France. [6]These authors contributed equally: Megha Roy, Aurore Sanchez. ✉e-mail: aurore.sanchez@curie.fr; petr.cejka@irb.usi.ch

MSH4-MSH5 (MutSγ), EXO1, and RFC-PCNA[10,15–19]. The involvement of PCNA is reminiscent of postreplicative mismatch repair, where PCNA guides the mismatch repair-specific MLH1-PMS2 (MutLα) endonuclease to cleave the newly replicated DNA strand marked by strand discontinuities[20,21]. It is thought that asymmetric retention of PCNA (loaded by RFC) on meiotic recombination intermediates could similarly guide the biased DNA cleavage by MutLγ[15,16,22]. MutSγ acts upstream of MutLγ to recognize Holliday junction intermediates[23], and thus functions as an analog of mismatch repair factors MSH2-MSH6 (MutSα) or MSH2-MSH3 (MutSβ) that recognize mismatches or insertion-deletion loops[24]. However, MutLγ also preferentially binds HJs[7]. The reason for the involvement of EXO1 in the pro-crossover meiotic pathway is much less apparent, as no role of EXO1 in promoting DNA cleavage by MutLα in mismatch repair has been found to date[24].

EXO1 is a multi-tasking double-stranded DNA (dsDNA) exonuclease with a 5′-3′ polarity. EXO1 functions in various steps of DNA metabolism including DNA end resection in homologous recombination and nucleolytic excision of the mismatched DNA strand in postreplicative mismatch repair[25–27]. In meiosis, EXO1 has at least two distinct roles[28,29]. First, EXO1 acts as a nuclease to facilitate the initial DNA end resection step. The resection function of EXO1 is apparent in yeast (exo1-deficient cells exhibit ~2-fold reduced resection length)[29,30], although the function of EXO1 nuclease activity in resection may be more redundant with other nucleases in mammalian meiosis[31–33]. Additionally, EXO1 plays a critical nuclease-independent role downstream together with MutLγ in crossover formation in both yeast and mammals[29,34,35]. In fact, mice expressing nuclease-deficient Exo1 are fertile, while mice entirely lacking Exo1 are sterile[35–37]. In yeast, Exo1 was also shown to protect nicks from ligation, which might explain its pro-crossover function[38]. Nuclease-deficient human recombinant EXO1 (D173A) was shown to promote the nicking activity of MutLγ, suggesting that it might act directly during the resolution step. In fact, EXO1 contains a MIP (MLH1-interaction protein sequence) motif that mediates a direct physical association with MLH1[29,39–41]. The integrity of the MIP motif is important, but not absolutely necessary for the pro-crossover function of Exo1 in yeast[29,38]. The role of the MIP motif in EXO1 in mammalian meiosis is not known, and how EXO1 promotes the meiotic MutLγ ensemble remains unclear. Here, we used purified recombinant proteins to define the role of EXO1 in the activation of the MutLγ nuclease complex together with MutSγ and RFC-PCNA. We show that the stimulatory effect of EXO1 on the MutLγ ensemble requires a three-pronged interaction with MLH1, MSH4 and DNA. Unexpectedly, disrupting the interaction of EXO1 with MLH1 has only a moderate effect on the MutLγ endonuclease. The identified interaction of EXO1 with MSH4 depends on a conserved EXO1 motif just downstream of its nuclease domain. A single point mutation in EXO1, W371E, fully disrupts its ability to promote the MutLγ ensemble, while it has no impact on the EXO1 nuclease activity. Our data support a model where EXO1, bound to dsDNA, is an integral structural part of the resolvase complex supported by conserved interactions with both MutSγ and MutLγ.

## Results

### EXO1 stimulates MutLγ only when MutSγ is present

Although magnesium ($Mg^{2+}$) is the most common nuclease metal cofactor in cells, it was observed in vitro that manganese ($Mn^{2+}$) can often replace $Mg^{2+}$ in the active site of many nucleases. While the enzymatic activity is often maintained, the presence of $Mn^{2+}$ may result in a decreased reaction specificity, as observed with many restriction endonucleases[42]. In the mismatch repair system, the canonical MutLα nuclease exhibits intrinsic nuclease activity in the presence of $Mn^{2+}$, without the requirement for any additional protein partners[20]. Its cofactors, including MutSα or MutSβ and RFC-PCNA are all required to observe nuclease activity in the presence of physiological

magnesium[20]. MutLγ (MLH1-MLH3) exhibits similar properties[15,16]. Incorporating $Mn^{2+}$ into the reconstituted reactions thus allows us to investigate MutLγ activation without the need for its complete set of partner proteins, helping us to gain insights into the activation mechanisms by the individual co-factors. We and the Hunter group have shown that the endonuclease activity of MutLγ (MLH1-MLH3) is stimulated by MutSγ (MSH4-MSH5), EXO1 (nuclease-deficient D173A variant was used), and RFC-PCNA[15,16]. In this study, we employed $Mn^{2+}$ to study the MutLγ nuclease activation by MutSγ and EXO1 without RFC-PCNA (Fig. 1a), and physiological $Mg^{2+}$ in the more complete reactions with MutSγ, EXO1, and also RFC-PCNA (Fig. 1a).

To study the activation of MutLγ by EXO1 and other co-factors, we used DNA nicking assays (Fig. 1a) and using negatively supercoiled dsDNA as a substrate and purified human recombinant proteins (Supplementary Fig. 1a). As previously[16], we observed that MutSγ, RFC-PCNA and EXO1 strongly promoted the nuclease activity of MutLγ (Fig. 1a). GLOE-seq analysis, which maps all 3′-OH ends with nucleotide resolution using next-generation sequencing, revealed that the DNA incision points are widely distributed on the plasmid DNA, although some sites are cleaved more efficiently than others (Fig. 1a). To find optimal conditions to observe the stimulation of the MutLγ ensemble by EXO1, we first varied the concentrations of $Mn^{2+}$/$Mg^{2+}$ (Fig. 1b, c, Supplementary Fig. 1b, c). We observed the greatest stimulatory effect of MutSγ-MutLγ by EXO1 (D173A) at low metal concentrations, generally between 0.3 and 1 mM $Mn^{2+}$ or $Mg^{2+}$ (Fig. 1b, c and Supplementary Fig. 1b, c). Under these conditions, the addition of EXO1 (D173A) stimulated the endonuclease of MutSγ-MutLγ up to ~5-fold, both without RFC-PCNA with $Mn^{2+}$ (Supplementary Fig. 1b, c) and with RFC-PCNA and $Mg^{2+}$ (Fig. 1b, c).

Importantly, we observed that EXO1 (D173A) stimulated the endonuclease activity of MutLγ only when MutSγ was present (Fig. 1d, e and Supplementary Fig. 1d)[16]. This observation applied for reactions both with and without RFC-PCNA (Fig. 1d). The requirement for the presence of MutSγ was apparent under both low and high $Mn^{2+}$/$Mg^{2+}$ concentrations (0.3 vs. 5 mM), showing that it is not due to a particular experimental condition (Fig. 1d, e and Supplementary Fig. 1d). The observed endonuclease activity of the ensemble reactions was intrinsic to MutLγ, as substituting wild-type MutLγ with the nuclease-deficient MutLγ-3ND (D1223N, Q1224K, E1229K) mutant largely eliminated DNA cutting (Fig. 1e and Supplementary Fig. 1d)[16]. The interaction of MutLγ with EXO1 in the meiotic pro-crossover pathway is known[29,34,39–41]. Our data revealed here an unexplored interplay between MutSγ and EXO1, which is the focus of this study.

### Physical interaction of EXO1 with MutL γ only partially contributes to its activation

Yeast Exo1 was described to physically and functionally interact with MutLγ through a conserved MLH1-interaction protein (MIP) motif in its C-terminal region[41]. Mutations in the MIP motif of yeast Exo1 impaired two-hybrid interaction with Mlh1[41]. The MIP motif mutation in yeast Exo1 consequently reduced crossing over during meiotic recombination, but not to the same extent as exo1Δ[29,38]. Little is known about the role of MLH1 and EXO1 interaction and its importance for meiotic recombination in humans. Human EXO1 contains the conserved MIP motif (R-S-R-F-F, residues 503–507, Fig. 2a)[41]. AlphaFold2 modeling[43] of EXO1 and MLH1 also predicted the interaction through the MIP motif of EXO1 and identified possible additional contact points involving residues 390–410 of EXO1 (Fig. 2a, b, Supplementary Fig. 2a). To define the importance of the interaction between MutLγ and EXO1 for the endonuclease activity of MutLγ, we constructed human EXO1 (D173A) variants with mutations in the MIP motif (F506A and F507A) (Fig. 2b, c). We also prepared the EXO1 (I403E) mutant targeting a conserved residue in the second possible interacting region, without or in combination with the MIP motif mutations (Fig. 2b, c). The proteins were purified (Supplementary Fig. 2b) and assayed for physical interaction

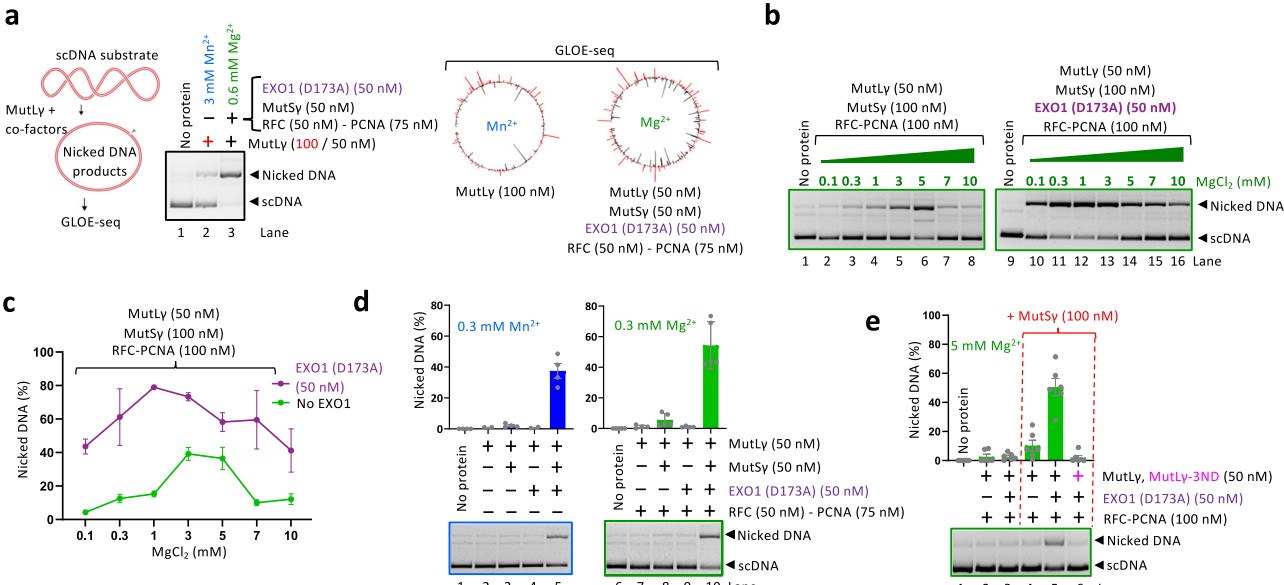

**Fig. 1 | EXO1 stimulates MutLγ only in the presence of MutSγ. a** A schematic of the nicking assay used to evaluate the endonuclease activity of MutLγ. Supercoiled DNA (scDNA) was incubated with MutLγ and its co-factors generating nicked DNA, which has different mobility in agarose gel electrophoresis (left). Right, polar plots of reads from GLOE-seq performed on scDNA (5.5 kbp). Briefly, after cleavage by the indicated proteins, all resulting 3′-OH ends were mapped by next-generation sequencing. The DNA cleavage sites in the top and bottom strand are in red and black, respectively. **b** Nicking assays with MutLγ, MutSγ and RFC-PCNA in the presence of increasing magnesium concentrations. The reaction was carried out in the absence (left) or presence (right) of nuclease-dead EXO1 (D173A). The separation was performed on a 1% agarose gel in the presence of GelRed. **c** Quantification of nicking assays such as in (**b**); averages shown. $n = 3$ independent experiments; error bars, SEM. **d** Nicking assays with indicated proteins either in the presence of 0.3 mM Mn$^{2+}$ and without RFC-PCNA (left), or in the presence of 0.3 mM Mg$^{2+}$ and RFC-PCNA (right). Top, quantification; averages shown, $n = 4$ (independent experiments with Mn$^{2+}$) and $n = 5$ (independent experiments with Mg$^{2+}$); error bars, SEM. Bottom, representative gels. **e** Nicking assays testing the effect of MutSγ and/ or EXO1 (D173A) on the stimulation of MutLγ at 5 mM Mg$^{2+}$ in the presence of RFC-PCNA. Nuclease-dead MutLγ (MutLγ-3ND, D1223N, Q1224K, E1229K) was used as a control. Top, quantification; averages shown, $n = 6$ independent experiments; error bars, SEM. Bottom, a representative gel. Source data are provided as a Source Data file.

with MutLγ. MutLγ was immobilized using anti-MLH1 antibody and incubated with the EXO1 variants (Fig. 2d, top). We observed that the MIP mutations in EXO1 did not significantly impair the physical interaction with MutLγ (Fig. 2d, Supplementary Fig. 2c). The I403E mutation notably reduced the physical interaction of the EXO1 MIP mutants with MutLγ, although it had no effect on its own (Fig. 2d, Supplementary Fig. 2c). We next tested the impact of these mutations on the nuclease activity of MutLγ. Quite unexpectedly, the MIP mutations in EXO1 (D173A) had no apparent effect on the endonuclease activity of MutLγ together with MutSγ, or with MutSγ and RFC-PCNA (Fig. 2e), similarly to the I403E mutation (Fig. 2e). Only the combination of the MIP and I403E mutations was moderately inhibitory, although the decrease in DNA cleavage was not statistically significant (Fig. 2e, lanes 5 and 12). In contrast, *S. cerevisiae* Exo1 (D173A), which is quite similar to the human protein in biochemical activities and primary structure (Fig. 2a), did not stimulate the nicking reaction by the MutSγ-MutLγ ensemble at all (Fig. 2e, lanes 7 and 14). Therefore, while the human EXO1 (D173A I403E MIP) variant is impaired in its physical interaction with MutLγ, it is still able to notably stimulate the MutSγ-MutLγ nuclease in a species-specific manner (Fig. 2e). It remains possible that the residual interaction between EXO1 (D173A I403E MIP) and MLH1 contributes to the nuclease activity of the ensemble. More likely, the experiments suggested that the integrity of the MLH1-EXO1 interface is partially dispensable for the nuclease activity of MutLγ, and that another unexplored interface within the MutSγ-MutLγ-EXO1 complex may also be important, as will be seen below.

## EXO1 directly physically interacts with MutSγ

The experiments above suggested a possible functional interaction between human EXO1 and MutSγ. To test whether the proteins directly physically interact, we carried out pulldown experiments with the

respective purified recombinant proteins. We first immobilized MutSγ (exploiting a his-tag on MSH5) and incubated it with EXO1 (D173A) (Fig. 3a, top). We could readily observe a direct interaction in pull-downs followed by western blotting (Fig. 3a). The interaction was unaffected by the presence of MutLγ (Fig. 3a), showing that MutLγ does not stabilize the ternary complex, but also that it does not interfere with the MutSγ-EXO1 interaction. The direct physical interaction was also revealed in experiments where MutSγ was immobilized via a Strep-tag on MSH4 (Fig. 3b), or reciprocally when EXO1 was immobilized exploiting its FLAG-tag (Fig. 3c). In contrast to the human proteins, the interaction between yeast Exo1 and yeast MutSγ was barely detectable under the same experimental conditions (Supplementary Fig. 3a), suggesting that the interaction may not be conserved in unicellular eukaryotes. We have also explored the protein interactions using mass photometry. EXO1 (D173A) was monomeric and MutSγ largely heterodimeric as expected (Fig. 3d). We then incubated the same concentrations of the proteins together and could observe species corresponding to the combined molecular weight of MutSγ and EXO1, albeit only a small proportion (~4% of total protein counts, at 50 nM protein concentration, Fig. 3d). The experiments confirmed the direct physical interaction between MutSγ and EXO1.

## EXO1 region 353−390 is required for MutSγ-MutLγ activation

We next set out to map the EXO1 region necessary for the activation of the endonuclease activity of MutLγ in conjunction with MutSγ. The N-terminal domain of EXO1 (residues 1−352) represents the catalytic nuclease domain with a determined crystal structure[44]. Downstream, the extended C-terminal domain is unstructured according to MobiDB[45,46] (Supplementary Fig. 3b) and AlphaFold2[47]. The C-terminal region of EXO1 mediates protein-protein interactions, including with MLH1 via the MIP box (residues 503−507) (Fig. 2b) and with PCNA via

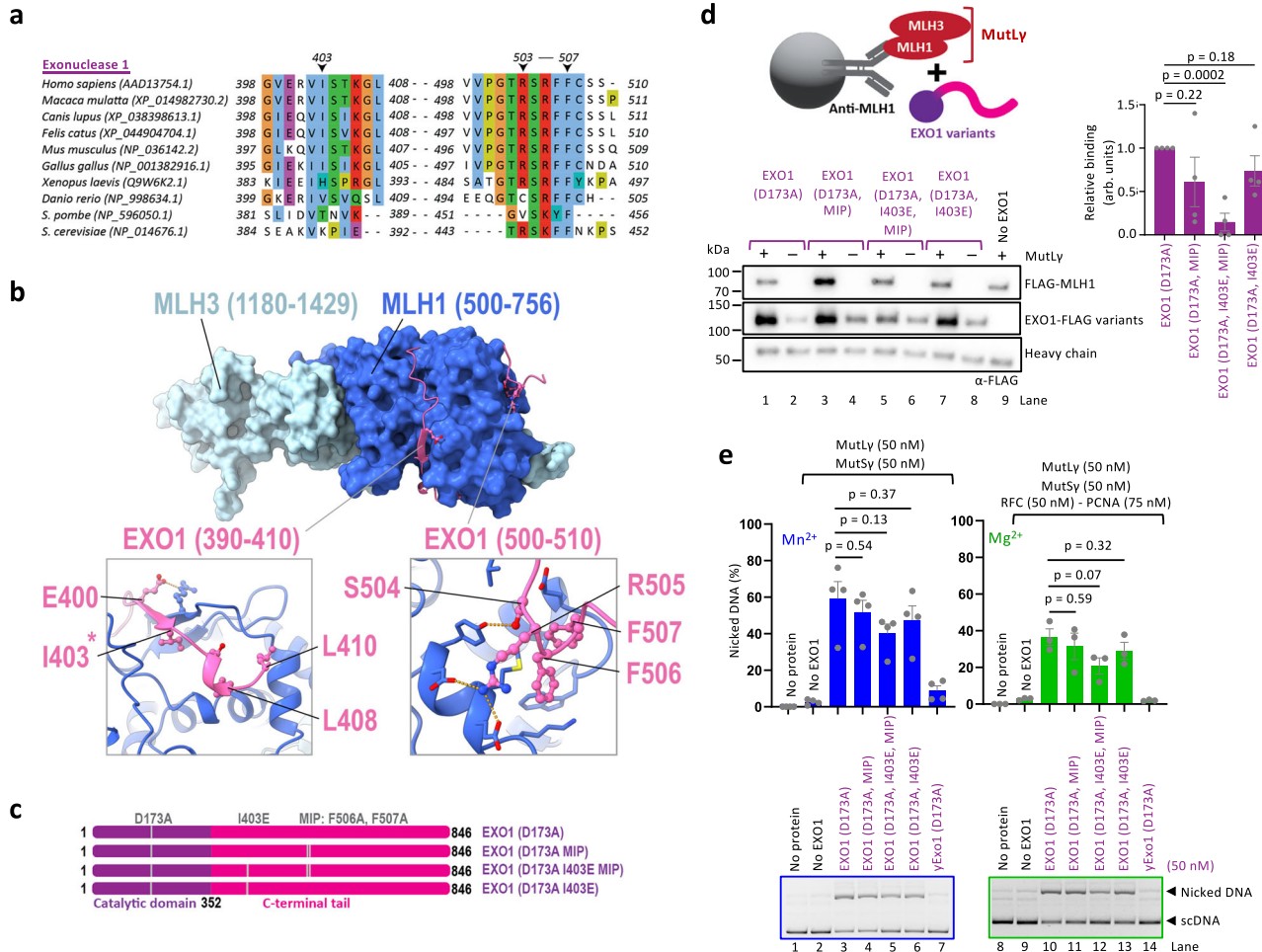

**Fig. 2 | Interaction of EXO1 with MLH1 has only a minor impact on the nuclease of MutSγ·MutLγ. a** Multiple sequence alignment of a conserved region of EXO1 from various species generated with MAFFT and visualized in Jalview using ClustalW color coding. Dashes (−) represent gaps in the alignment. **b** AlphaFold2 model of the complex between the C-terminal domains of MutLγ and EXO1. Top, C-terminal domains of MutLγ are shown in light and dark blue for MLH3 and MLH1, respectively, and the interacting regions of EXO1 are shown in pink. Bottom left, residues interacting with MLH1 in the 390–410 stretch of EXO1 (centered around I403). Bottom right, conserved MIP motif (R-S-R-F-F, residues 503–507) of EXO1 interacting with MLH1. Asterisk indicates the residue mutated in this study. **c** Cartoons of the primary structure of EXO1. The N-terminal catalytic domain is shown in purple, and the C-terminal tail is shown in magenta. Mutations of EXO1 residues predicted to be involved in the interaction with MutLγ are highlighted. MIP mutations, F506A and F507A mutations within the MIP motif. The nuclease-dead

D173A mutation is indicated for reference. **d** Protein interaction assays of MutLγ and EXO1 variants. MutLγ (FLAG-MLH1 and MBP-MLH3, bait) was immobilized using an anti-MLH1 antibody, and EXO1-FLAG variants (prey) were subsequently added. MIP mutations, F506A and F507A mutations within the MIP motif. Top, a schematic. Bottom, a representative experiment. Right, quantification of interaction relative to EXO1 (D173A), normalized to MLH1, averages shown, $n = 4$ independent experiments; error bars, SEM; two-tailed $t$-test. **e** Nicking assays testing the effect of EXO1 (D173A) variants on the stimulation of MutLγ-MutSγ complex in the presence of 0.6 mM $Mn^{2+}$ (without RFC-PCNA) (left); and in the presence of 0.3 mM $Mg^{2+}$ and RFC-PCNA (right). MIP mutations are F506A and F507A within the MIP motif. Nuclease-dead yeast Exo1 (D173A) was used in lanes 7 and 14 as a negative control. Top, quantification; averages shown, $n = 4$ (experiments with $Mn^{2+}$) and $n = 3$ (experiments with $Mg^{2+}$); error bars, SEM; two-tailed $t$-test. Bottom, representative gels. Source data are provided as a Source Data file.

the PIP box (PCNA-interacting protein, residues 788–795) (Fig. 3e). The C-terminal region also contains numerous phosphorylation sites with unknown functions, and a nuclear localization signal (residues 418–421). We next created a series of EXO1 truncation mutants within the nuclease-deficient D173A background, each lacking consecutively larger segments of the C-terminal region (Fig. 3e, f). The mutants were then subject to nicking assays either with MutSγ and MutLγ in $Mn^{2+}$, or additionally with RFC-PCNA in $Mg^{2+}$ (Fig. 3g). We observed only a minimal and statistically not significant reduction of the endonuclease activity when truncating the C-terminal domain of EXO1 up to residue 390, consistent with the unexpectedly minor role of the MLH1-interaction sites that map to these regions (Fig. 2), as well as the PIP motif as determined previously (Fig. 3e)[16,48]. However, when we truncated EXO1 even further, down to residue 352, we observed almost a complete drop in the EXO1 variant's capacity to stimulate MutSγ-MutLγ, irrespective of the presence of RFC-PCNA (Fig. 3g). The

catalytic domain of EXO1, corresponding to residues 1–352, did not stimulate MutLγ at all, and was thus indistinguishable from yeast full-length Exo1 (D173A) (Fig. 3g, compare lanes 8 and 9, and 17 and 18). Therefore, residues 353–390 of human EXO1 are necessary for the species-specific activation of the MutSγ-MutLγ nuclease complex.

We next assayed for the physical interaction between the EXO1 variants and MSH4-5. MutSγ was immobilized using anti-His antibody and incubated with the EXO1 variant (Fig. 3h, top). We observed a robust interaction of MutSγ with full-length EXO1 and EXO1 1-390, while the interaction with EXO1 1-352 was dramatically reduced (Fig. 3h, Supplementary Fig. 3c). The EXO1 variants were FLAG-tagged at the C-terminus, and we note that the anti-FLAG antibody detected the variants similarly (Supplementary Fig. 3c). Finally, we analyzed the dsDNA binding capacity of the EXO1 (D173A) truncation mutants in electrophoretic mobility shift assays. The primary DNA-binding activity of EXO1 is likely dependent on its N-terminal nuclease domain[44]. We

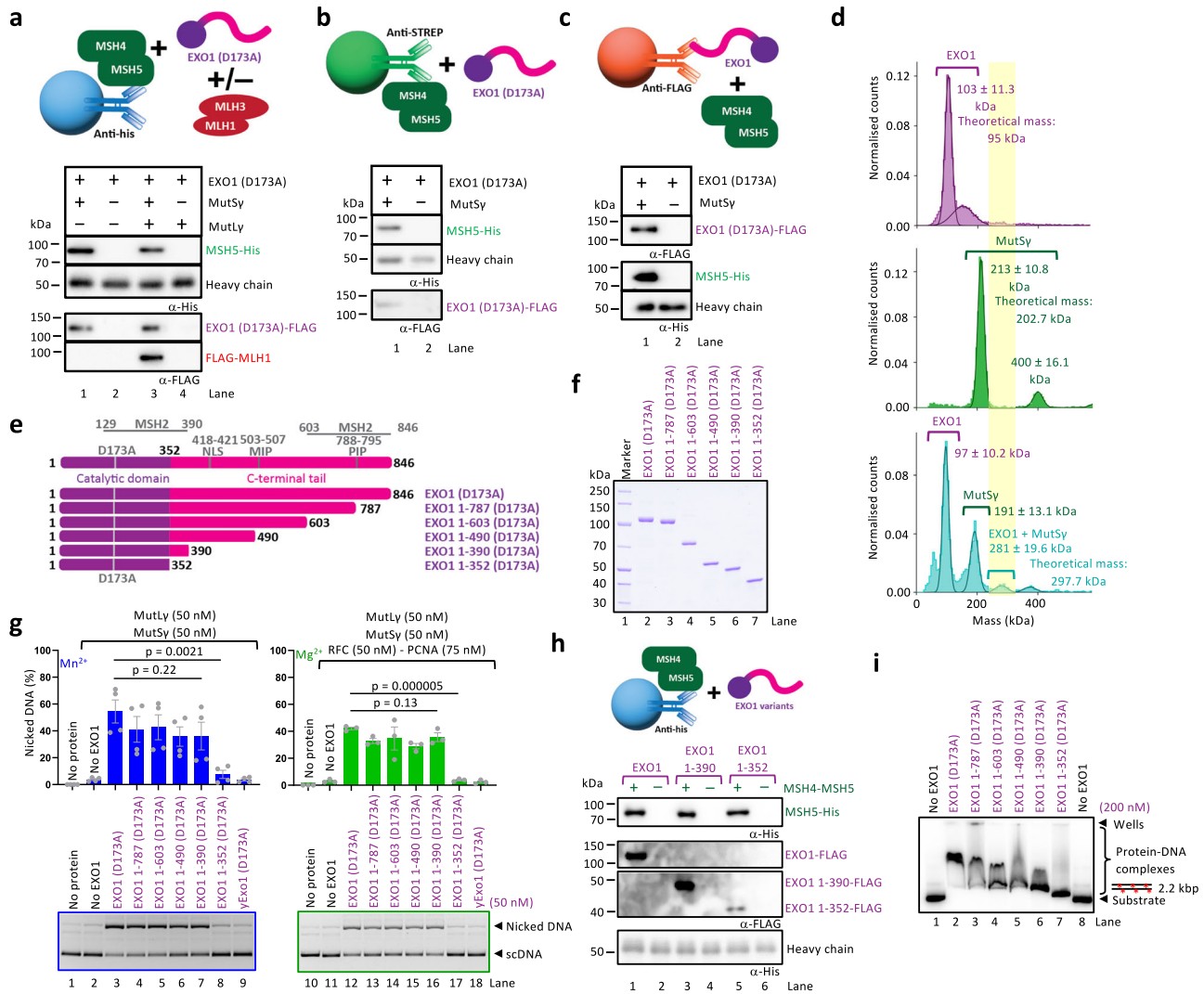

**Fig. 3 | Physical and functional interaction of EXO1 with MutSγ depends on residues 353–390 in EXO1. a** Protein interaction assays of MutSγ and EXO1 (D173A) with or without MutLγ. MutSγ (MSH4-Strep-MSH5-his) (bait) was immobilized using an anti-His antibody, and EXO1 (D173A)-FLAG and MutLγ (FLAG-MLH1-MBP-MLH3) (preys) were subsequently added. Top, a schematic. Bottom, a representative of two independent experiments. **b** Protein interaction assay of MutSγ and EXO1 (D173A). MutSγ (MSH4-Strep-MSH5-His) (bait) was immobilized using Strep resin and EXO1 (D173A)-FLAG (prey) was subsequently added. Top, a schematic. Bottom, a representative of three Western blot. **c** Protein interaction assay of MutSγ and EXO1 (D173A). EXO1 (D173A)-FLAG (bait) was immobilized using anti-FLAG resin and MutSγ (MSH4-Strep-MSH5-His) (prey) was subsequently added. Top, a schematic. Bottom, a representative of two independent experiments. **d** Molecular weight distributions of EXO1 (D173A)-FLAG and/or MutSγ (MSH4-Strep-MSH5-His) measured by mass photometry. Error, SD. **e** A schematic overview of EXO1 (D173A) protein variants. The N-terminal catalytic domain is shown in purple, and the C-terminal tail is shown in magenta. Top, EXO1 protein with important functional

regions highlighted. NLS nuclear localization signal, MIP motif MLH1 interacting protein-box, PIP motif PCNA-interacting protein-box, EXO1 regions interacting with MSH2. Bottom, a cartoon of EXO1 truncations. **f** A representative of two polyacrylamide gel showing purified recombinant EXO1 (D173A) truncation mutants. **g** Stimulation of MutLγ-MutSγ complex nicking activity by EXO1 (D173A) truncation mutants in the presence of 0.3 mM $Mn^{2+}$ (left) and in the presence of 0.3 mM $Mg^{2+}$ and RFC-PCNA (right). Nuclease-dead yeast Exo1 (D173A) was used as a negative control. Top, quantification; averages shown, $n = 4$ (experiments with $Mn^{2+}$) and $n = 3$ (experiments with $Mg^{2+}$); error bars, SEM; two-tailed t-test. Bottom, representative gels. **h** Protein interaction assays of MutSγ and EXO1 variants. MutSγ (MSH4-Strep-MSH5-His) (bait) was immobilized using an anti-His antibody, and EXO1-FLAG, EXO1 1-390-FLAG, or EXO1 1-352-FLAG (preys) were subsequently added. Top, a schematic. Bottom, a representative of three independent experiments. **i** Electrophoretic mobility shift assays with EXO1 (D173A) truncation mutants. Red asterisks, radioactive labels. A representative of three independent experiments is shown. Source data are provided as a Source Data file.

observed a gradual increase in the mobility of shifted DNA corresponding to the extent of the C-terminal truncation, suggesting that the C-terminus of EXO1 may also contribute to DNA binding (see also below). However, the reduced protein-DNA mobility shift may also in part correspond to the decreasing EXO1 protein size (Fig. 3i). Nevertheless, EXO1 1-390 (D173A) and EXO1 1-352 (D173A) still retain DNA binding capacity, (Fig. 3i), suggesting that DNA binding alone is unlikely to explain the difference between the stimulatory capacities of two respective EXO1 truncation mutants. We conclude that EXO1 and MutSγ physically and functionally interact, and that the interaction

depends on the C-terminal unstructured region of EXO1, between residues 353–390 downstream of the nuclease domain.

## Residues 353–390 in EXO1 do not affect the nuclease activity of EXO1

To better understand the role of the 353–390 residues of EXO1 in the activation of MutSγ-MutLγ, we also prepared wild-type full-length EXO1, as well as the EXO1 1-390 and EXO1 1-352 truncation variants in the nuclease-proficient background (Fig. 4a, b). As shown in Fig. 4c, all three EXO1 variants exhibited a similar exonuclease activity on

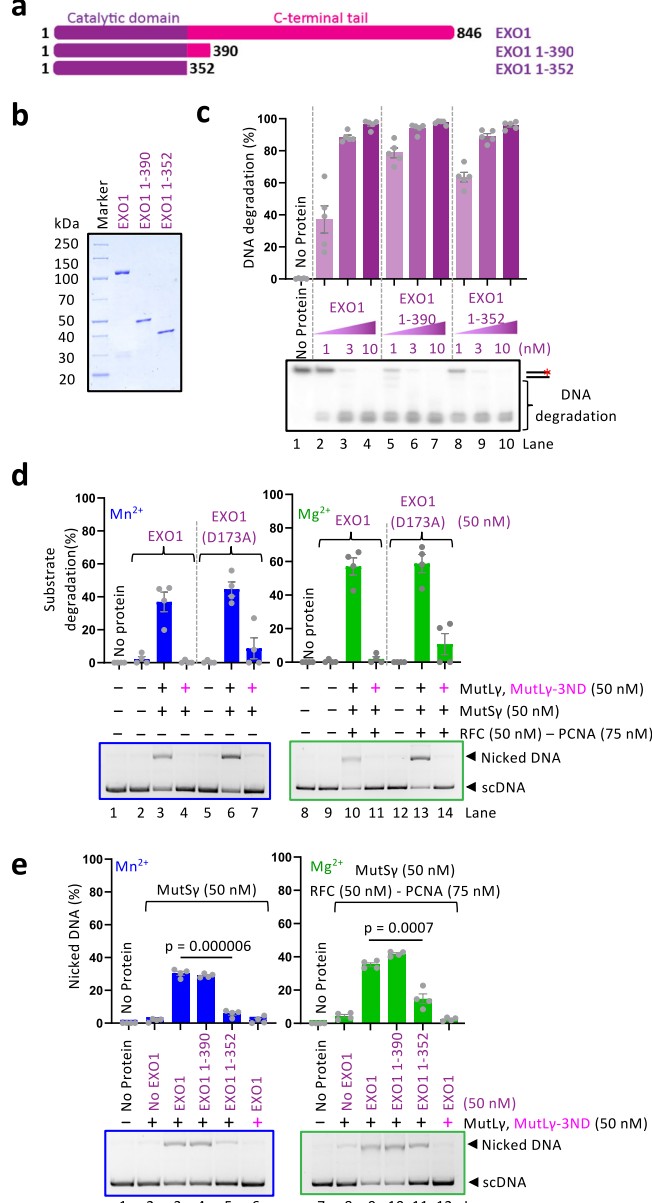

oligonucleotide-based dsDNA (in 5 mM $Mg^{2+}$), demonstrating that the 353–390 region of EXO1 (and even the whole C-terminal domain) is dispensable for the intrinsic nuclease activity. Compared to wild-type full-length EXO1, the truncations exhibited notably reduced apparent electrophoretic mobility in DNA binding assays (Supplementary Fig. 4a), which is likely due to their reduced size compared to full-length EXO1 and a contribution of the C-terminal EXO1 region to DNA binding (see below). Both constructs remained monomeric, showing no signs of aggregation in mass photometry (Supplementary Fig. 4b).

We observed that full-length wild-type and nuclease-deficient EXO1 (D173A) variants promoted DNA incisions by MutSγ-MutLγ with the same efficacy based on substrate DNA utilization (Fig. 4d). The detectable amount of nicked DNA product was lower with wild-type EXO1, because a fraction of the nicked DNA was degraded by the exonuclease activity of EXO1 (Fig. 4d). These experiments showed that the nuclease activity of EXO1 is dispensable for the stimulation of the MutSγ-MutLγ nuclease, in accord with genetic studies in yeast that showed that the nuclease activity of Exo1 is dispensable for dHJ resolution[29]. We note that the nuclease activity of EXO1 is partially suppressed by the low $Mg^{2+}$/$Mn^{2+}$ concentration used (0.6 mM, Fig. 4d).

Similarly to the data obtained with the nuclease-deficient EXO1 (D173A) variants, we also observed that full-length wild-type EXO1 and EXO1 1-390 stimulated the incision by MutSγ-MutLγ, while EXO1 1-352 was incapable to do so, either in the presence or in the absence of RFC-PCNA (Fig. 4e). These experiments demonstrated that the 353–390 region of EXO1 is uniquely important for the stimulation of MutSγ-MutLγ, but dispensable for the nuclease activity of EXO1, showing that the two EXO1 functions—intrinsic nuclease activity and the stimulation of MutSγ-MutLγ nuclease ensemble—are genetically separable.

## The integrity of W371 of EXO1 is necessary for the stimulation of MutSγ-MutLγ

In vertebrates, the EXO1 region encompassing amino acids 353 to 390 contains a cluster of conserved residues at positions 369 to 375 (Fig. 5a). Using a domain scanning strategy to increase the sensitivity of the detection[49], AlphaFold2 modeling identified a potential interaction between this region of EXO1 and the N-terminal domain of MSH4 (Fig. 5a, Supplementary Fig. 5a). We first set out to test the model by making key point mutations on the EXO1 side. We constructed W371E and Y375E (EE) double as well as S369E, W371E and Y375E (EEE) triple mutant within the full-length nuclease-deficient EXO1 (D173A) background (Fig. 5c, d). We observed that both mutant combinations abolished the capacity of EXO1 (D173A) to promote the nicking activity by MutSγ-MutLγ and RFC-PCNA (Fig. 5e). Similar results were obtained using the nuclease-proficient EXO1 variants. The EXO1 (EE) and EXO1 (EEE) mutants, both within the full-length protein and within the 1–390 fragment were incapable of stimulating MutSγ-MutLγ (Fig. 5f, g, Supplementary Fig. 5b, c), while they were proficient in DNA binding (Fig. 5h, Supplementary Fig. 5d), retained their intrinsic nuclease activity (Fig. 5i, Supplementary Fig. 5e), and did not aggregate (Supplementary Fig. 5f). The EXO1 (EE) mutant thus clearly separates the various functions of EXO1, affecting only the stimulation of the MutLγ nuclease ensemble.

To further narrow down the critical residues in EXO1 required to stimulate MutSγ-MutLγ, we mutated the two residues, W371 and Y375, individually into negatively charged glutamic acid residues (W371E and Y375E) and individually or combined into neutral alanines (W371A and Y375A) (Fig. 5f, g). We observed that both substitutions of W371 were disruptive, while the mutagenesis of Y375 had little impact (Fig. 5j, k). This is consistent with the anchoring role of W371 in the structural model, while Y375 is more peripheral (Fig. 5b). These experiments identified the W371 of EXO1 as the key residue, the integrity of which is required to stimulate MutSγ-MutLγ (Fig. 5j, k).

**Fig. 4 | Stimulation of MutSγ-MutLγ by nuclease-proficient EXO1 variants. a** A schematic overview of catalytically active EXO1 full-length and truncation variants used in this study. The N-terminal catalytic domain is shown in purple, and the C-terminal tail is shown in magenta. **b** Recombinant catalytically active full-length EXO1 and truncation mutants used in this study. The polyacrylamide gel was stained with Coomassie brilliant blue. A representative of two independent experiments. **c** Nuclease assays using 3′-labeled 50 bp dsDNA with indicated EXO1 variants. Top, quantification. Averages shown, $n = 5$ independent experiments; error bars, SEM. Bottom, a representative gel. The red asterisk indicates the position of the radioactive label. **d** Stimulation of MutLγ-MutSγ complex nicking activity by wild-type EXO1 or nuclease-dead EXO1 (D173A). Reactions were carried out in the presence of 0.6 mM $Mn^{2+}$ and in the absence of RFC-PCNA (left) or in the presence of 0.6 mM $Mg^{2+}$ and RFC-PCNA (right). Nuclease-dead MutLγ (MutLγ-3ND, [D1223N, Q1224K, E1229K]) was used as a control (lanes 4, 7, 11 and 14). Top, quantification; averages shown, $n = 4$ for experiments in $Mn^{2+}$ and $n = 7$ for experiments in $Mg^{2+}$; error bars, SEM. Bottom, representative gels. **e** Stimulation of MutLγ-MutSγ complex nicking activity by nuclease-proficient full-length EXO1, EXO1 1-390 and EXO 1-352. Reactions were carried out in the presence of 0.6 mM $Mn^{2+}$ in the absence of RFC-PCNA (left) or in the presence of 0.6 mM $Mg^{2+}$ and RFC-PCNA (right). Nuclease-dead MutLγ (MutLγ-3ND, [D1223N, Q1224K, E1229K]) was used as a control (lanes 6 and 12). Top, quantification; averages shown, $n = 4$ independent experiments; error bars, SEM. Bottom, representative gels, two-tailed $t$-test. Source data are provided as a Source Data file.

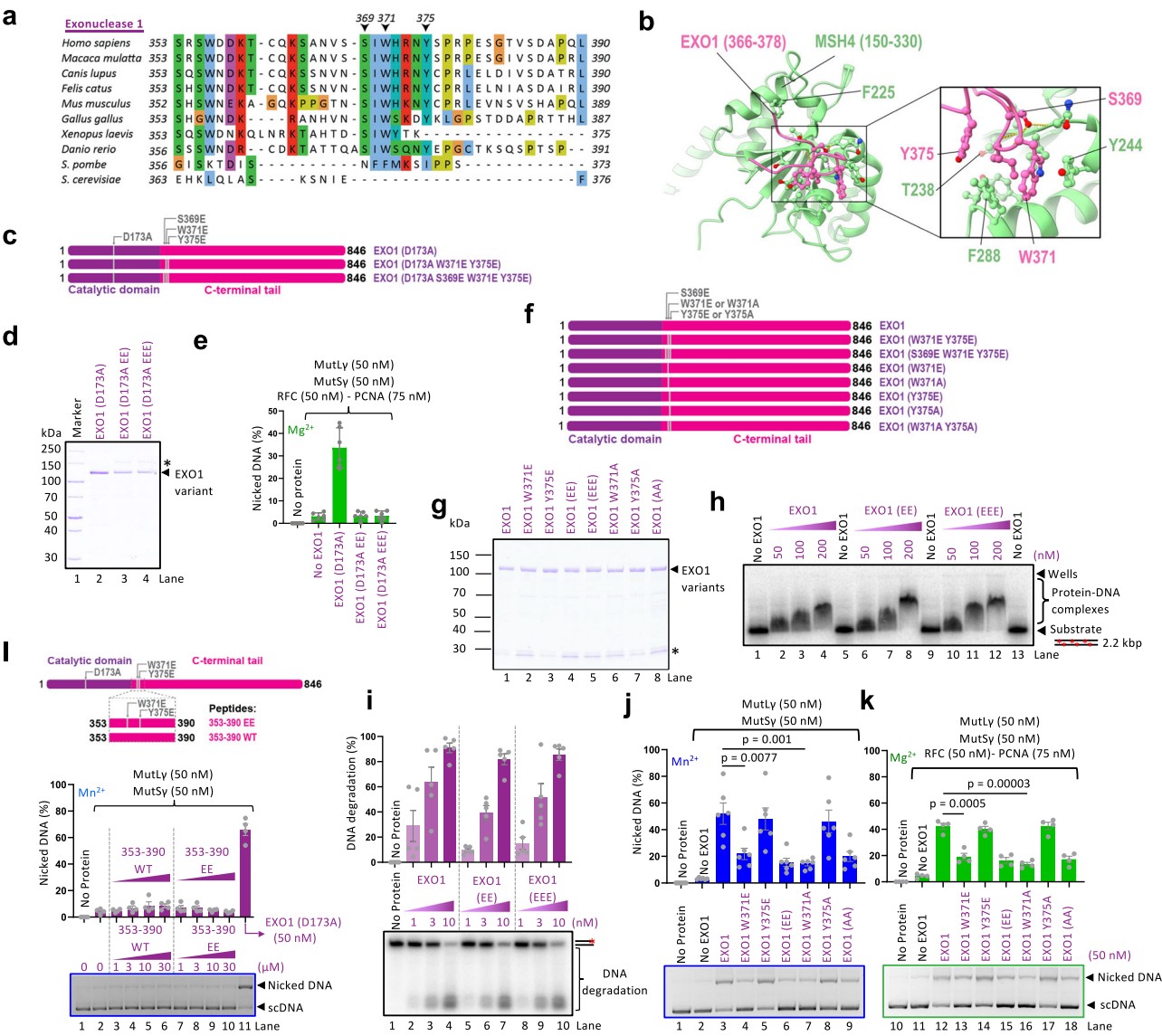

**Fig. 5 | Residues around W371 of EXO1 are critical for MutSγ-MutLγ stimulation.** **a** Multiple sequence alignment of EXO1 region spanning residues 353–390. Dashes (−) represent gaps in the alignment. **b** AlphaFold2 model of the interaction between the N-terminal region of MSH4 and EXO1. Left, N-terminal domains of MSH4 (light green) and the interacting regions of EXO1 (residues 366–378) (pink). Right, predicted interacting residues. **c** A schematic of EXO1 (D173A) variants used in this study. EXO1 (D173A W371E Y375E) is abbreviated as EXO1 (D173A EE) and EXO1 (D173A S369E W371E Y375E) is abbreviated as EXO1 (D173A EEE). **d** A representative of two polyacrylamide gel showing purified EXO1 point mutants. Asterisk (*) indicates uncleaved MBP-tagged EXO1 variants. **e** Nicking assays of MutLγ-MutSγ complex with EXO1 (D173A) variants at 0.3 mM Mg$^{2+}$ in the presence of RFC-PCNA. Averages shown, $n = 6$ independent experiments; error bars, SEM. **f** A schematic of EXO1 point mutants. EXO1 (W371E Y375E) is abbreviated as EXO1 (EE), EXO1 (S369E W371E Y375E) is abbreviated as EXO1 (EEE) and EXO1 (W371A Y375A) is abbreviated as EXO1 (AA). **g** A representative of two polyacrylamide gel showing EXO1 mutants.

Asterisk (*) indicates a non-specific band. **h** Electrophoretic mobility shift assays with recombinant EXO1 variants. Red asterisks, radioactive labels. A representative of three independent experiments. **i** Nuclease assay with EXO1 variants. Top, quantification. Averages shown, $n = 5$ independent experiments; error bars, SEM. Bottom, representative gel. Red asterisk, radioactive label. **j** Stimulation of MutLγ-MutSγ complex nicking activity by EXO1 variants in the presence of 0.6 mM Mn$^{2+}$. Top, quantification; averages shown, $n = 6$ independent experiments; error bars, SEM. Bottom, representative gels; two-tailed $t$-test. **k** Experiment as in (**j**) with 0.6 mM Mg$^{2+}$ and RFC-PCNA. Top, quantification; averages shown, $n = 4$ independent experiments; error bars, SEM; two-tailed $t$-test. Bottom, representative gels. **l** The nicking activity of the MutLγ-MutSγ complex with EXO1 WT or EE (W371E Y375E) peptides in the presence of 0.6 mM Mn$^{2+}$. Top, a cartoon. Middle, quantification; averages shown, $n = 4$ independent experiments; error bars, SEM. Bottom, representative gel. Source data are provided as a Source Data file.

The above experiments showed that the 353–390 region of EXO1 is necessary to stimulate MutSγ-MutLγ. To test whether it is sufficient, we used a peptide encompassing the 353–390 region either with the wild-type sequence or with the EE mutations (Fig. 5l, top). We observed that neither of the two peptides could substitute full-length EXO1 (D173A) (Fig. 5l). The structured N-terminal nuclease domain upstream of the 353–390 region in EXO1 is thus required for MutSγ-MutLγ activation.

## The EXO1 cluster around W371 interacts with the N-terminal domain of MSH4

The AlphaFold2 model predicted the conserved EXO1 cluster to interact with residues in the N-terminal part of the MSH4 subunit of MutSγ (Fig. 5b) defined as the connector domain in the MutS protein family[50]. To test the structural model on the MSH4 side, we created four combinations of point mutants targeting the surface exposed and conserved residues predicted to be most likely involved in the

interaction with EXO1 (Fig. 6a). All MSH4 mutants could be purified as heterodimers with MSH5 (Supplementary Fig. 6a). The mutated MutSγ complexes were all dimeric with 1:1 stoichiometry as determined by mass photometry (Supplementary Fig. 6b). All MutSγ variants were able to stimulate the nuclease activity of MutLγ (without EXO1), showing that also the interplay with MutLγ is intact (Fig. 6b). However, the triple MutSγ mutations (T238E, Y244D, F288E) clearly impaired the nuclease activity of MutLγ in the presence of EXO1 (D173A), irrespectively of RFC-PCNA (Fig. 6c). We next assayed for the physical interaction between the MutSγ variants and EXO1. EXO1 was immobilized using anti-FLAG antibody and incubated with the MutSγ variants (Fig. 6d, top). We observed that the physical interaction of EXO1 with the MutSγ variants was reduced, mostly so with the MutSγ triple mutant (Fig. 6d). These data support the Alpha-Fold2 structural model (Fig. 5b), where Y244 in MSH4 is predicted to directly contact W371 in EXO1. Supplementary Fig. 6c shows that the modeled interface between MSH4 and EXO1 in other vertebrate species, such as *Xenopus laevis* and *Danio rerio*, is predicted by AlphaFold2 to adopt a similar conformation. Interestingly, the region of MSH4 involved in binding with EXO1 exhibits a distinct conserved pocket primarily composed of hydrophobic residues. This contrasts with other homologs of the MutS family, which also have conserved positions in this region but include at least two charged residues (Supplementary Fig. 6d). In summary, the conserved physical interaction between EXO1 and the MSH4 subunit of MutSγ is needed for the stimulation of the MutLγ nuclease.

### DNA binding, but not nuclease activity, of the N-terminal EXO1 domain promotes MutSγ-MutLγ

Our experiments suggested that the N-terminal domain of EXO1 (residues 1–352), in addition to the 353–390 region described above, is necessary for the stimulation of MutSγ-MutLγ (Fig. 5l). We note that EXO1 truncations from the N-terminal side were unstable and could not be purified. A previous study in yeast cells found that the DNA binding activity of Exo1, dependent on the N-terminal domain, is required for meiotic recombination[38]. Based on genetic experiments, the authors proposed that Exo1 protects MutLγ-dependent nicks from ligation[38]. The biochemical data presented in previous reports[15,16] and here demonstrate that human EXO1 is an integral component of the endonuclease ensemble together with MutSγ-MutLγ. Therefore, the function of human EXO1 in promoting meiotic recombination seems at least in part different from the yeast homolog, supported by the lack of conservation of the interaction motif between EXO1 and MSH4 (Fig. 5a). We next set out to test for the role of the DNA binding activity of human EXO1 on the stimulation of the MutSγ-MutLγ nuclease ensemble.

The crystal structure of human EXO1 identified two metals in its active site[44]. It is thought that one $Mg^{2+}$ is coordinated by residues including D78, while the other $Mg^{2+}$ is coordinated by D173 (Fig. 7a)[44]. The key aspartate residues are exactly at the same numerical positions in both yeast and human proteins, showing a high level of conservation. The D78A and D173A mutations disrupt the nuclease activity of yeast Exo1, and had no impact on meiotic recombination as scored by genetic assays[38]. The crystal structure of EXO1 also revealed two lysine residues located away from the $Mg^{2+}$ ions in close contact with DNA, and correspondingly K185D and K237D mutations (corresponding to K185E and G236D in yeast) are predicted to disrupt DNA binding (Fig. 7a)[44].

To define the role of $Mg^{2+}$ coordination and DNA binding by human EXO1 in the stimulation of MutSγ-MutLγ, we constructed EXO1 (D78A) and EXO1 (K185D, K237D) mutations both within the full-length protein and the EXO1 1-390 fragment (Fig. 7b). As anticipated, the D78A mutation completely disrupted the nuclease activity of EXO1, similarly to the D173A mutation, both within the full-length and the truncated constructs (Fig. 7c). The $Mg^{2+}$ coordination site

mutants instead did not affect DNA binding (Fig. 7d). In contrast, the K185D K237D mutations only partially reduced the nuclease activity of full-length EXO1, although they had a much stronger impact in the 1-390 EXO1 fragment (Fig. 7c). Furthermore, while the K185D K237D mutations eliminated detectable DNA binding activity of the short 1-390 EXO1 variant, the mutations had only a minimal impact on the DNA binding activity of full-length EXO1 both in electrophoretic mobility shift assays (Fig. 7d) and in experiments where biotinylated dsDNA was used as a bait for EXO1 variant binding (Fig. 7e)[44]. These data agree with the analysis of the EXO1 truncation mutants (Fig. 3i), which suggested that the C-terminal domain of EXO1 contributes to DNA binding.

We next set out to define the capacity of the N-terminal EXO1 point mutants to stimulate the endonuclease activity of MutSγ-MutLγ. The EXO1 (D78A) mutant behaved indistinguishably from the EXO1 (D173A) mutant, showing that $Mg^{2+}$ coordination and nuclease activities of EXO1 are dispensable for the stimulation of MutSγ-MutLγ (Fig. 7f), in accord with the analysis of the corresponding mutants in yeast meiotic recombination[44]. In contrast, the K185D K237D mutations significantly reduced the capacity of the tested EXO1 variants to promote MutSγ-MutLγ, most notably of the EXO1 1-390 variant that lacks the additional DNA binding regions locating to the C-terminal part of EXO1 (Fig. 7f). We note that the K185D K237D in EXO1 instead did not affect its interaction with MutSγ or MutLγ (Supplementary Fig. 7a, b) and did not aggregate as determined by mass photometry (Supplementary Fig. 7c). We conclude that DNA binding by the N-terminal domain of EXO1, mediated by K185 and K237 residues, and to a lesser extent by the C-terminal domain of EXO1, contributes to its structural function to stimulate the nuclease activity of the MutSγ-MutLγ-RFC-PCNA ensemble. In contrast, metal-coordinating residues of EXO1 necessary for its intrinsic nuclease function are dispensable for the nicking activity of the MutSγ-MutLγ-RFC-PCNA ensemble. Previous studies indicated that MutSγ and MutLγ bind preferentially to HJs and show only a weak affinity for dsDNA. We observed that EXO1 binds in contrast dsDNA quite efficiently (Supplementary Fig. 7d, e), in accord with previous work. These data collectively suggest that EXO1 might tether the MutSγ and MutLγ proteins to dsDNA, which might explain its structural role in promoting DNA cleavage.

## Discussion

Meiotic recombination intermediates are resolved specifically into crossovers by a conserved pathway dependent on the MutLγ endonuclease[4–12]. We have previously demonstrated[16], along with the Hunter laboratory[15], that the endonuclease activity of the MutLγ complex is directly stimulated by MutSγ, EXO1 and RFC-PCNA. Our studies suggested that MutSγ and EXO1 are co-factors of the MutLγ endonuclease ensemble and proposed that PCNA might direct the biased cleavage of meiotic recombination intermediates into crossovers. The model evoked parallels with the postreplicative mismatch repair machinery, where PCNA directs MutLα to cleave the nascent DNA strand marked by strand discontinuities[15,22]. The role of EXO1 in the MutLγ nuclease complex remained undefined. It is apparent from both genetic studies and our biochemical reconstitution assays that EXO1 promotes the MutLγ endonuclease ensemble independently of its nuclease activity[15,16,29,34,35]. The involvement of EXO1 in the activation of MutLγ hence fundamentally differs from its role in MMR. In MMR, EXO1 does not promote the incision by MutLα, but rather acts as an exonuclease during the subsequent degradation of the nascent DNA strand containing the mismatched residues[20].

Here, we used a reconstituted system developed previously[16] to define the role of EXO1 in the activation of the meiotic MutLγ endonuclease ensemble (Fig. 1). The primary structure of EXO1 contains a conserved nuclease domain (residues 1–352)[44], and an unstructured C-terminal region up to the last residue at position 846.

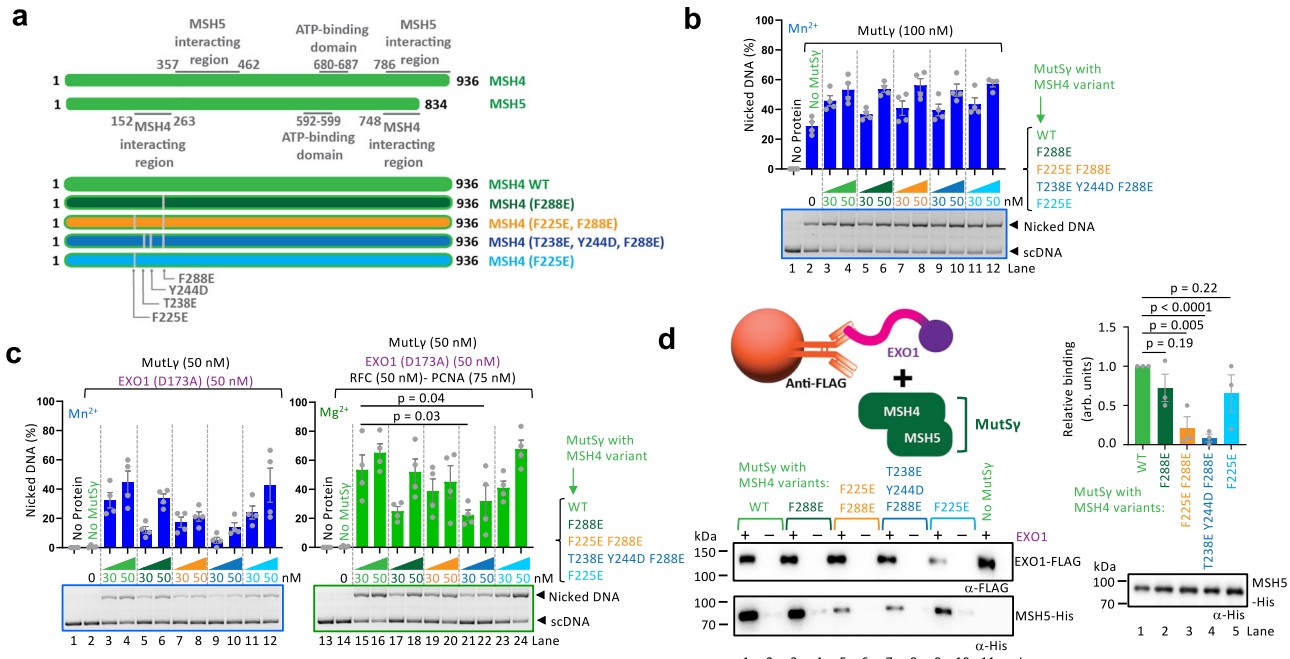

**Fig. 6 | Residues around Y244 in MSH4 are critical for the interaction between EXO1 and MutSγ-MutLγ. a** Top, a schematic of the primary structure of MutSγ (MSH4-MSH5), with key features indicated. Bottom, a cartoon of predicted MSH4 point mutants affecting interaction with EXO1 used in this study (see also Fig. 5b). **b** Stimulation of MutLγ nicking activity by the indicated MutSγ mutants in the presence of 3 mM Mn²⁺ without any EXO1 variant. Top, averages shown, $n = 4$ independent experiments; error bars, SEM. Bottom, a representative gel. **c** Stimulation of MutLγ nicking activity by the indicated MutSγ variants in the presence of EXO1 (D173A) and either 0.6 mM Mn²⁺ without RFC-PCNA (left) or 0.6 mM Mg²⁺ with RFC-PCNA (right). Top, quantification; averages shown, $n = 4$

independent experiments; error bars, SEM; two-tailed $t$-test. Bottom, representative gels. **d** Left, protein interaction assays of MutSγ and EXO1. EXO1-FLAG (bait) was immobilized using anti-FLAG antibody, and MutSγ variants (MSH4-Strep-MSH5-His) (prey) were subsequently added. A schematic of the assay (top), a representative experiment analyzed by Western blotting (bottom). Right top, quantification of the interaction relative to MutSγ (WT), normalized to EXO1; averages shown, $n = 3$ independent experiments; error bars, SEM; two-tailed $t$-test (top). Right bottom, Western blot of the MutSγ variants. 100 ng of the recombinant proteins were loaded and detected using anti-HIS antibody. Source data are provided as a Source Data file.

---

EXO1 is known to interact with MLH1 via a conserved MIP motif (residues 503–507), the integrity of which is important for crossover formation in yeast[29,39–41]. Using AlphaFold2 modeling, we identified additional contact points with MLH1 mediated by residues 400–410 of EXO1. Disrupting both sites in EXO1 strongly reduced the physical interaction with MutLγ, but it did not have a significant impact on the stimulation of the MutLγ endonuclease ensemble (Fig. 2). We noted that EXO1 stimulated MutLγ, but only when MutSγ was present[16] (Fig. 1). We observed that EXO1 directly interacts with MutSγ (Fig. 3), which involves residues in EXO1 immediately downstream of the nuclease domain (369 to 375, the key residue being W371) (Fig. 4), modeled to interact with the MSH4 subunit around Y244 (Fig. 5). In contrast to disrupting the interaction sites with MLH1, mutations in the MSH4 interaction patch of EXO1 completely abolished the ability of EXO1 to promote MutLγ-MutSγ, independently of RFC-PCNA. The intrinsic nuclease function of EXO1 is mutationally fully separable from its capacity to stimulate MutLγ-MutSγ. Mutations in EXO1 that disrupt magnesium coordination and thus abolish the EXO1 nuclease activity (D173A, D78A) have no effect on the stimulation of MutLγ-MutSγ (Fig. 7). In contrast, a single point mutation in EXO1, W371E, eliminates the stimulation of MutLγ-MutSγ, but does not affect the nuclease activity of EXO1 (Fig. 5). The AlphaFold2 modeling and biochemical analysis suggested that intrinsically disordered regions in EXO1 and MSH4 act as molecular tethers that facilitate the assembly of the MutLγ partners on DNA for optimal activation of its nuclease activity.

A recent study in yeast demonstrated that residues K185 and G236, involved in DNA binding, are critical for crossover formation[38]. It

was proposed that the DNA binding activity of Exo1 prevents the ligation of nicks on meiotic recombination intermediates, which facilitates their maturation into crossover product[38]. We observed here that the corresponding mutations in human EXO1 (K185D, K237D) strongly inhibit the ability of EXO1 to promote MutLγ-MutSγ (Fig. 7). Our data thus support a different model in the human system. We conclude that DNA binding by EXO1 is pre-requisite for it to act as a direct structural component of the human MutLγ-MutSγ meiotic nuclease ensemble. Our data obtained using the human recombinant proteins are nevertheless not in conflict with the nick-protection function proposed in yeast. The meiotic defects of yeast *exo1* and *mlh1*-deficient strains are similar, while notable differences exist in mice, suggesting that the regulation and the interplay of EXO1 and MutLγ are likely different between yeast and mammals[34]. Additionally, the EXO1 and MSH4 interaction motifs identified here are conserved in vertebrates but not in low eukaryotes (Fig. 5a).

MutLγ and MutSγ proteins bind HJs, but have only a weak affinity for dsDNA[7,23]. It was proposed that MutLγ is initially recruited to HJs, but it is thought to cleave dsDNA some distance away from the junctions[12]. Our data demonstrate that the structural function of EXO1 to stimulate the MutSγ-MutLγ endonuclease ensemble marginally depends on its interaction with MLH1 of MutLγ, and then importantly on its interaction with MSH4 of MutSγ and its DNA binding activity. A possible model is that after MutSγ and MutLγ recognize and are activated by HJs, they may slide along the dsDNA arms, and EXO1 functions as a molecular staple to keep MutSγ and MutLγ tethered to dsDNA. Activated MutLγ then cleaves dsDNA on meiotic recombination intermediates, ultimately resulting in crossovers. Our study also has

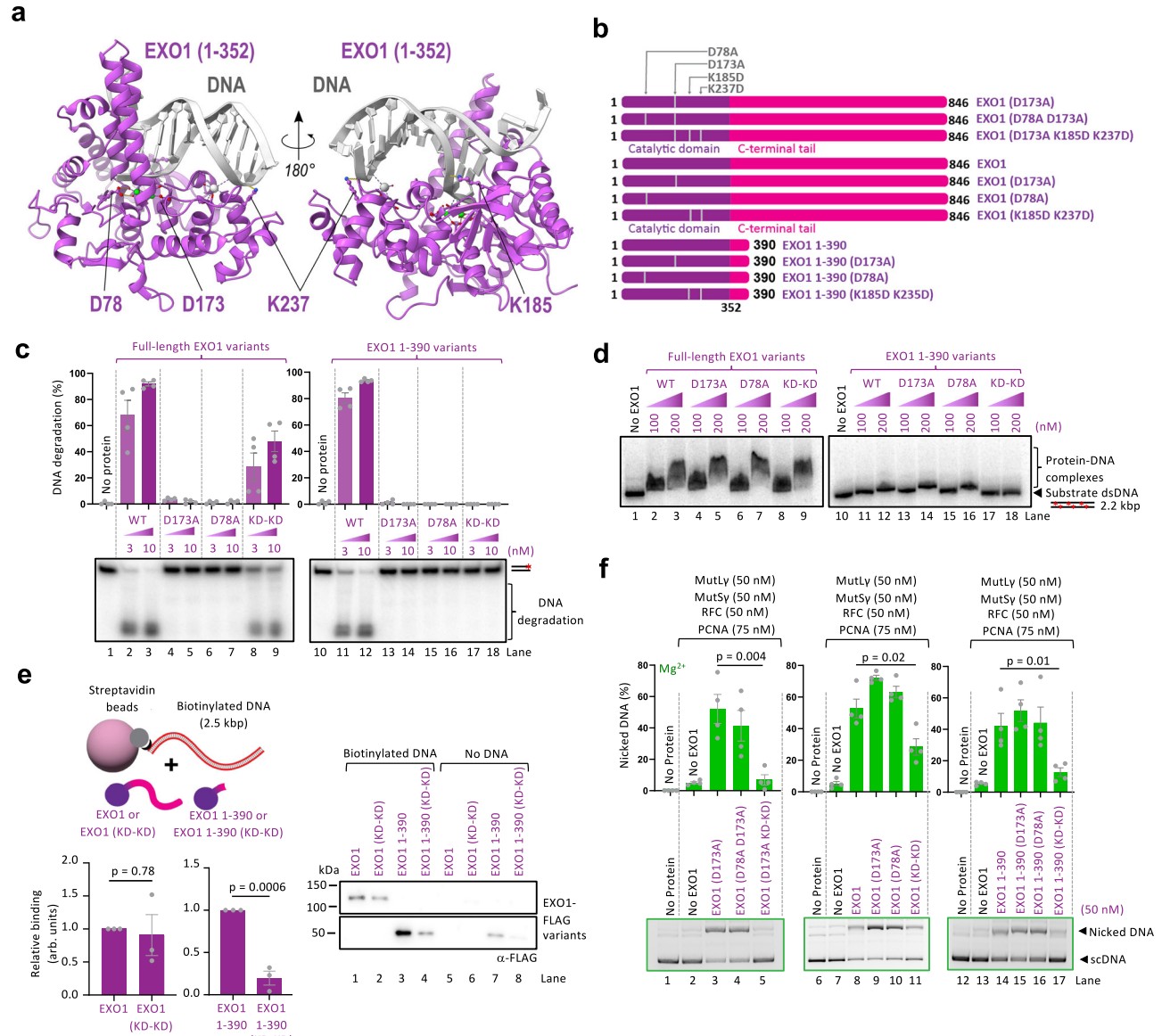

**Fig. 7 | DNA binding by EXO1 is necessary for MutLγ-MutSγ stimulation.**
**a** Interaction of EXO1 with DNA from the previously reported crystal structure[44]. Left, residues D78 and D173 in the EXO1 catalytic domain (shown in purple) coordinate Mg$^{2+}$ (shown in green) essential for its catalytic activity. Right, residues K185 and K237 of EXO1 are in close contact with DNA (shown in gray), with a potential role in DNA binding. **b** A cartoon of full-length EXO1 or EXO1 1-390 mutants used in this study. The mutation K185D K237D is abbreviated as KD-KD hereafter. **c** Nuclease assays using 3'-labeled 50 bp dsDNA with indicated full-length EXO1 and EXO1 1-390 variants. Top, quantification. Averages shown, $n = 4$ independent experiments; error bars, SEM. Bottom, a representative gel. The red asterisk indicates the position of the radioactive label. **d** Electrophoretic mobility shift assays with recombinant full-length EXO1 and EXO1 1-390 variants as indicated, using randomly labeled 2.2 kbp-long dsDNA. The red asterisks indicate the radioactive labels. Shown is a representative of two independent experiments. **e** Interaction assay of DNA and EXO1 variants. Biotinylated DNA (2.5 kbp) (bait) was immobilized on streptavidin beads, and EXO1 variants (preys) were subsequently added. Left, schematic of the assay (top), quantification of the interaction relative to either EXO1 or EXO1 1-390; averages shown, $n = 3$ independent experiments; error bars, SEM; two-tailed $t$-test (bottom). Right, a representative experiment analyzed by Western blotting. **f** Stimulation of MutLγ-MutSγ complex nicking activity by the indicated EXO1 variants in the presence of 0.6 mM Mg$^{2+}$ and RFC-PCNA. Top, quantification; averages shown, $n = 4$ independent experiments; error bars, SEM; two-tailed $t$-test. Bottom, representative gels. Source data are provided as a Source Data file.

potential implications for understanding the molecular basis of certain infertility cases resulting from mutations in these meiotic recombination factors.

## Methods

### Protein expression in insect cells
MutLγ, MutSγ, EXO1 and RFC proteins were purified from *Spodoptera frugiperda* (*Sf*9) cells using the Bac-to-Bac expression system (Invitrogen) following the manufacturer's instructions. Subsequently, the cells were collected, washed with 1x phosphate-buffered saline (PBS), frozen in liquid nitrogen, and stored at −80 °C until purification. All the purification steps were performed at 4 °C.

### Purification of human MutLγ (MLH1-MLH3)
Human MutLγ (MLH1-MLH3) was expressed in *Sf*9 cells using recombinant baculoviruses made from pFB-FLAG-hMLH1co and pFB-His-MBP-hMLH3co vectors. The protein complex was purified through affinity purification using amylose resin (New England Biolabs) exploiting maltose-binding protein (MBP) tag on MLH3, followed by M2 anti-FLAG affinity resin (Sigma) exploiting FLAG-tag on MLH1. MBP-

tag was cleaved using PreScission Protease (PP)[16]. Nuclease-dead MutLγ-3ND (D1223N, Q1224K and E1229K mutations in MLH3) was generated using the QuikChange II XL site-directed mutagenesis kit, following the manufacturer's instructions (Agilent Technology). Sequences of all primers used for site-directed mutagenesis are listed in Supplementary Table 1. The protein was expressed and purified following the same procedure as the wild-type protein[16].

## Purification of yeast and human MutLγ (MSH4-MSH5)

Human MutSγ (MSH4-MSH5) was expressed from a dual pFBDM-hMSH4co-STREP-hMSH5co-His vector in *Sf*9 cells, and purified by NiNTA (Qiagen) and STREP (Qiagen) affinity purification steps[16]. Mutations in *MSH4* were generated using the QuikChange II XL site-directed mutagenesis kit, following the manufacturer's instructions (Agilent Technology). Sequences of all primers used for site-directed mutagenesis are listed in Supplementary Table 1. The variants of the MSH4-MSH5 complex were purified using the same procedure[16]. Yeast MutSγ (Msh4-Msh5) was expressed in *Sf*9 cells using recombinant baculoviruses made from pFB-yMSH4-STREP and pFB-yMSH5-His vectors, and purified by NiNTA and STREP affinity purification[16].

## Purification of yeast and human EXO1

Nuclease-deficient human EXO1 variants (bearing the D173A mutation) were prepared from pFB-MBP-PP-EXO1(D173A)-1xFLAG construct and its derivatives. To prepare pFB-MBP-PP-EXO1(D173A)-1xFLAG, the sequence coding for EXO1 (D173A) was amplified from pFB-EXO1-FLAG vector by PCR using primers HEXO1FO_NHE (5′-GCGCGCGCTAGCATGGGGATACAGGGATTGC-3′) and HEXO1RE (5′-CGCGCGCCCGGGCTACTTGTCGTCATCGTCTTTGTAGTCCTGGAA-TATTGCTCTTTGAAC-3′)[16]. The PCR product was digested by NheI and XmaI (New England Biolabs) and cloned into corresponding sites in pFB-MBP-Sae2-His[51]; the sequence of *SAE2* was removed during the process, and His-tag from the original construct was not translated due to a stop codon after the 1xFLAG tag. *EXO1* point mutations were created using the QuikChange II XL site-directed mutagenesis kit according to the manufacturer's instructions (Agilent Technologies). To create *EXO1* truncation variants, segments of *EXO1* (D173A) were amplified by PCR using the forward primer HEXO1FO_NHE and the respective reverse primer; AS21 for EXO1 (D173A) 1-787, AS22 for EXO1 (D173A) 1-603, AS23 for EXO1 (D173A) 1-490, AS33 for EXO1 (D173A) 1-390 and AS34 for EXO1 (D173A) 1-352, and cloned into pFB-MBP-Sae2-His as stated above. The sequences of primers used are listed in Supplementary Table 1.

All human EXO1 (D173A) variants were expressed in *Sf*9 cells. The cell pellet was resuspended in three volumes of lysis buffer (50 mM Tris-HCl pH 7.5, 1 mM β-mercaptoethanol, 1 mM ethylenediaminetetraacetic acid [EDTA], 1:400 protease inhibitor cocktail [Sigma, P8340], 0.5 mM phenylmethylsulphonyl fluoride [PMSF], 20 μg/ml leupeptin [Merck-Millipore]). Then, one-half volume of 50% glycerol and 6.5% volume of 5 M NaCl (final concentration 305 mM) were added to the cell suspension and incubated on a roller for an additional 30 min, followed by centrifugation at 55,000×*g*. The soluble extract was added to pre-equilibrated amylose resin (New England Biolabs) and incubated for 45 min. After incubation, the soluble extract-resin suspension was centrifuged at 2000×*g* and the supernatant (flow-through) was removed. The resin was washed with amylose wash buffer-1 (25 mM Tris-HCl pH 7.5, 1 mM β-mercaptoethanol, 1 M NaCl, 10% glycerol and 0.5 mM PMSF). Then, the resin was transferred to a disposable 10 ml polypropylene column (Thermo Fisher Scientific) and further washed with amylose wash buffer-1 followed by amylose wash buffer-2 (25 mM Tris-HCl pH 7.5, 1 mM β-mercaptoethanol, 300 mM NaCl, 10% glycerol, 0.5 mM PMSF and 0.5 mM EDTA). The protein was eluted with MBP wash buffer-2 supplemented with 10 mM maltose (Sigma), and the total protein concentration was estimated by the Bradford assay. To cleave the maltose-binding protein (MBP) tag, one-

fourth (weight/weight) of PreScission Protease (PP) was added to the amylose eluate and incubated for 60 min. Next, the PP-cleaved eluate was diluted by adding a half-volume of FLAG dilution buffer (25 mM Tris-HCl, 10% glycerol, 0.5 mM PMSF and 0.2% NP-40) to lower the concentration of β-mercaptoethanol. The solution was then applied to pre-equilibrated FLAG M2 affinity resin in 5 ml polypropylene column (Thermo Fisher Scientific) in flow. The resin was first washed with FLAG wash buffer-1 (25 mM Tris-HCl, 1 mM β-mercaptoethanol, 150 mM NaCl, 10% glycerol, 0.5 mM PMSF, 0.2% NP-40) followed by FLAG wash buffer-2 (25 mM Tris-HCl, 1 mM β-mercaptoethanol, 150 mM NaCl, 10% glycerol). Finally, the protein was eluted with FLAG wash buffer-2 supplemented with 150 ng/μl 3xFLAG peptide (Sigma), aliquoted and frozen in liquid nitrogen, and stored at −80 °C. The final protein contained a 1xFLAG tag at the C-terminus of EXO1. All EXO1 (D173A) variants were purified using the same procedure.

Nuclease-proficient human EXO1 variants were expressed in *Sf*9 cells from a pFB-EXO1-FLAG vector. To create the EXO1 truncation variants EXO1 1-390 and EXO1 1-352 without MBP, the respective EXO1 segments were amplified using primers M1 and M2 and cloned into pFB-MBP-PP-EXO1(D173A)-1xFLAG using BamHI and XmaI, and D173A mutation was reverted back to WT, resulting in pFB-EXO1-1-390-1xFLAG and pFB-EXO1-1-352-1xFLAG. Mutations in the expression vectors were created using the QuikChange II XL site-directed mutagenesis kit, following the manufacturer's instructions (Agilent Technology). Sequences of all primers used for site-directed mutagenesis are listed in Supplementary Table 1. The cell pellet was resuspended in three pellet volumes of lysis buffer (25 mM Tris-HCl pH 7.5, 0.5 mM β-mercaptoethanol, 1 mM EDTA, 1:400 protease inhibitor cocktail, 0.5 mM PMSF, 20 μg/ml leupeptin). The cell suspension was incubated with gentle stirring for 10 min. One-half volume of 50% glycerol and 6.5% volume of 5 M NaCl (final concentration 305 mM) were added. The suspension was incubated for 30 min with stirring. The extract was then centrifuged at 55,000×*g* for 30 min. The soluble extract was added to pre-equilibrated M2 anti-FLAG affinity resin (Sigma) and incubated for 45 min. The suspension was then centrifuged (2000×*g*), the supernatant (FLAG flow-through) removed, and the resin was transferred to a disposable chromatography column. The resin was washed with wash buffer (20 mM Tris-HCl pH 7.5, 150 mM NaCl, 0.5 mM β-mercaptoethanol, 0.5 mM PMSF, 10% glycerol) supplemented with 0.1% NP40. This step was followed by washing with wash buffer without NP40. EXO1-FLAG was eluted with wash buffer supplemented with 150 ng/μl 3×FLAG peptide and stored at −80 °C after snap freezing in liquid nitrogen. The final construct contained a FLAG tag at the C-terminus of EXO1. All nuclease-proficient EXO1 variants were purified using the same procedure.

Yeast Exo1 (D173A) was purified from *Sf*9 cells using the pFB-Exo1(D173A)-FLAG vector through FLAG affinity and HiTrap SP HP (Cytiva) ion exchange chromatography[52].

## Purification of human and yeast RFC

Human RFC was purified by infecting *Sf*9 cells with recombinant baculovirus prepared using the pFBDM-MBP-RFC1-RFC2-3-4-His-5 vector[16] (a kind gift from Josef Jiricny, ETH Zurich). The RFC1 subunit was MBP-tagged while the RFC5 subunit was His-tagged. The complex was purified using NiNTA and MBP affinity chromatography, followed by ion exchange chromatography using a HiTrap Q HP column (Cytiva)[16]. Yeast RFC was expressed in *E. coli* cells using the pEAO271 plasmid (a kind gift from E. Alani, Cornell University). The protein was purified using the HiTrap SP column (Cytiva) followed by the HiTrap Q column[16].

## Purification of human PCNA

Human PCNA was expressed from the pET23C-his-hPCNA vector (a kind gift from Ulrich Hübscher, University of Zürich) in *E. coli* cells and

purified using NiNTA resin followed by HiTrap Q HP column chromatography[16,53].

## Preparation of DNA substrates

The 50 bp-long dsDNA was prepared by annealing the oligonucleotides X12-3 (5′-GACGTCATAGACGATTACATTGCTAGGACATGCTGTCTAGA-GACTATCGC-3′) and X12-4C (5′ GCGATAGTCTCTAGACAGCATGTCC TAGCAATGTAATCGTCTATGACGTC-3′). HJ was prepared by annealing the oligonucleotides PC1253 (5′- TGGGTCAACGTGGGCAAAGATGT CCTAGCAATGTAATCGTCTATGACGTT-3′), PC1254 (5′-TGCCGAATTC-TACCAGTGCCAGTGATGGACATCTTTGCCCACGTTGA CCC-3′), PC1255 (5′-GTCGGATCCTCTAGACAGCTCCATGATCACTGGCACTGGTAGAATT CGGC-3′) and PC1256 (5′-CAACGTCATAGACGATTACATTGCTACATGG AGCTGTCTAGAGGATCCGA-3′). X12-3 and PC1253 were [32]P-labeled at the -3′ terminus using (α-[32]P)dCTP (Hartmann Analytic) and terminal transferase (New England Biolabs) according to the manufacturer's instructions. Unincorporated (α-[32]P)dCTP was removed using the Micro Bio-Spin P-30 Tris chromatography columns (BioRad). The 2.2 kbp substrate was randomly labeled by amplifying the human *NBS1* gene in a PCR reaction containing 66 nM (α-[32]P) dCTP (Hartmann Analytic) along with the standard concentration of dNTPs (200 μM each)[54]. The PCR product was purified using QIAquick PCR & Gel Cleanup kit (Qiagen) and Chroma Spin TE-200 columns (Clonetech). The 2.5 kbp biotinylated dsDNA substrate was prepared by amplifying a region of pUC19 using primers NHEJ-FOR-BIO and NHEJ-REV-Nhe1 in a 50 μl PCR reaction containing 50 ng linearized pUC19, 1X HF buffer, 0.5 μM of each primer, 200 μM dNTPs and 0.3 unit of Phusion polymerase. Following PCR, the amplified DNA fragment was gel-purified using agarose gel electrophoresis and extracted with a Gel Extraction Kit (Qiagen).

## DNA nicking assays

The reactions (15 μl) were carried out in a solution containing 25 mM Tris-acetate pH 7.5, 1 mM dithiothreitol (DTT), 0.1 mg/ml bovine serum albumin (BSA, New England Biolabs), with either manganese chloride (MnCl₂, Sigma) or magnesium chloride (MgCl₂, Sigma) at specified concentrations. Additionally, ATP (2 mM, GE Healthcare, 27-1006-01) and plasmid-based DNA substrate (100 ng per reaction, 5.6-kb-long pFB-RPA2) were included. The reaction buffer was assembled, and the specified recombinant proteins (or peptides (Genecust), where indicated) were added on ice (MutLγ was always added last). The reactions were incubated for 60 min at 37 °C and stopped with 5 μl STOP solution (150 mM EDTA, 2% SDS, 30% glycerol, 0.01% bromophenol blue) and 1 μl proteinase K (Roche, 18 mg/ml) and further incubated for 60 min at 50 °C. The reaction products were then separated by electrophoresis in 1% agarose gel (Sigma, A9539) containing GelRed (Biotium) in TAE buffer. Subsequently, the gels were imaged (Quantum CX5 system). The results were quantitated using ImageJ and, unless indicated otherwise, expressed as a percentage of nicked DNA versus the total DNA in each lane. Any nicked DNA present in control (no protein) reactions was subtracted as background[16].

## Exonuclease assays

DNA exonuclease assays with EXO1 and its variants were performed in 15 μl volume in buffer containing 25 mM Tris-HCl pH 7.5, 5 mM MgCl₂, 1 mM DTT, 0.25 mg/ml BSA, 1 mM ATP, 1 mM phosphoenolpyruvate, 80 U/ml pyruvate kinase and 1 nM substrate (50 bp-long dsDNA, in molecules). The reaction was mixed, and the recombinant proteins were added on ice. The reaction was incubated for 30 min at 37 °C and later stopped by adding 1 μl of Proteinase K (Roche, 18 mg/ml) and 0.5 μl of 0.5 M EDTA for 30 min at 50 °C. An equal volume of formamide dye (95% [v/v] formamide, 20 mM EDTA, bromophenol blue) was added, samples were heated at 95 °C for 4 min, centrifuged (16,000×g) at 4 °C and kept briefly on ice. The products were then separated on 15% denaturing polyacrylamide gels (ratio acrylamide:-bisacrylamide 19:1 [Bio-Rad]). The gels were fixed in a solution

containing 40% methanol, 10% acetic acid and 5% glycerol for 30 min, dried on 3MM paper (Whatman), exposed to storage phosphor screens (GE Healthcare) and scanned by Phosphor imager (Typhoon FLA 9500, GE Healthcare). The data were analyzed using ImageJ and plotted with GraphPad Prism 10.

## Interaction assays

To examine the interaction between MutLγ and tested EXO1 variants, 0.6 μg anti-MLH1 antibody (Abcam, ab223844, clone number EPR20522) was immobilized on 10 μl protein G magnetic beads (Dynabeads, Invitrogen). To investigate the interaction between MutSγ and EXO1 variants, 1 μg anti-His antibody (MBL D291-3, clone OGHis) was captured on 10 μl protein G magnetic beads (His-tag is on the C-terminus of MSH5). To investigate the interaction between EXO1 and MutSγ variants, 1 μg anti-FLAG antibody (Sigma, F3165, Lot #SLCC4005) was captured on 10 μl protein G magnetic beads (FLAG-tag is on the C-terminus of EXO1). To investigate the interaction between MutSγ and EXO1 variants reciprocally, 1 μg anti-STREP antibody (BioRad, MCA2489, clone Strep-tag II) was captured on 10 μl protein G magnetic beads (STREP-tag is on the C-terminus of MSH4). The antibodies were incubated in 50 μl PBS-T (PBS containing 0.1% Tween-20) for 60 min at 4 °C with gentle mixing at regular intervals. The beads were then combined with 1 μg each recombinant bait (MutLγ, MutSγ or EXO1) in 100 μl binding buffer (25 mM Tris-HCl pH 7.5, 3 mM EDTA, 1 mM DTT, 20 μg/ml BSA, 50 mM NaCl) and incubated at 4 °C for 60 min with gentle rotation. After incubation, the beads were washed three times with 200 μl of wash buffer (50 mM Tris-HCl pH 7.5, 3 mM EDTA, 1 mM DTT, 100 mM NaCl, 0.05% Triton X-100 [Sigma]). 1 μg of the prey (EXO1 variants, MutSγ variants or MutLγ) was then added to the beads in 200 μl binding buffer and incubated at 4 °C for 1 h with gentle rotation. In the case of EXO1 truncation variants, equimolar quantities (100 nM) of the variants were added in 100 μl binding buffer. Beads were again washed 3 times with 200 μl of wash buffer, and the proteins were eluted by boiling the beads in SDS buffer (50 mM Tris-HCl pH 6.8, 1.6% sodium dodecyl sulfate, 1 mM DTT, 10% glycerol, 0.01% bromophenol blue) for 3 min at 95 °C. Avidin (Sigma) was added to the eluate as a stabilizer (0.1 μg/μl). To examine the DNA binding of EXO1 variants, 1 nM biotinylated DNA (2.5 kbp) and 200 nM of EXO1 variants were mixed in 20 μl DNA binding buffer (20 mM Tris-HCl pH 7.5, 75 mM NaCl, 0.1% Triton X-100, 3 mM EDTA, 1 mM DTT and 0.1 mg/mL BSA) and incubated for 15 min on ice. Next, 15 μl of streptavidin beads (Dynabeads M-280 Streptavidin, Invitrogen) was added to the reaction mixture and incubated at 4 °C with gentle rotation. The beads were then washed three times with 150 μl of DNA binding buffer (without BSA) and eluted by boiling in SDS buffer. Finally, 1 μg of BSA was added to the eluate as a stabilizer. The eluate was separated on a 7.5% SDS-PAGE gel, and the proteins were detected by Western blotting with anti-FLAG rabbit (Sigma, F7425, 1:1000) to detect both EXO1 and MutLγ, with anti-STREP mouse antibody (BioRad, MCA2489, clone Strep-tag II, dilution 1:1000) to detect MSH4 in MutSγ and with anti-His mouse antibody (MBL, D291-3, 1:5000) to detect MSH5 of MutSγ. The final images were acquired with FusionFX7 capture software (Vilber Imaging).

## Electrophoretic mobility shift assays

The EXO1-DNA binding reactions were carried out in 15 μl volume in a buffer containing 25 mM Tris-acetate pH 7.5, 1 mM DTT, 3 mM EDTA, 0.1 mg/ml BSA, 1 nM DNA substrate (in molecules, 2.2 kbp) and the indicated concentrations of recombinant proteins (EXO1 and their variants). For super-shift assays comprising EXO1 (D173A), MutLγ and MutSγ, the reactions were carried out in 15 μl volume in buffer containing 25 mM Tris-acetate pH 7.5, 1 mM DTT, 5% (volume/volume) glycerol, 5 mM MgCl₂, 50 μg/mL BSA, 50 μM ATP, 3.3 ng/μl dsDNA as competitor (50 bp), 0.5 nM DNA substrate and respective concentrations of recombinant proteins. The reactions were assembled and incubated on ice for 15 min, followed by the addition of 5 μl EMSA

loading dye (50% glycerol, 0.01% bromophenol blue). The products were separated on 0.6% agarose gels in TAE buffer at 4 °C. The gels were dried on DE81 paper (Whatman), exposed to storage phosphor screens (GE Healthcare) and scanned by Phosphorimager (Typhoon FLA 9500, GE Healthcare).

## Mass photometric characterization of protein complexes

Mass photometry measurements were conducted using a TwoMP mass photometer (Refeyn Ltd). Borosilicate microscope glass coverslips (No. 1.5 H thickness, 24 × 50 mm, VWR) were cleaned by soaking them sequentially in Milli-Q-water, isopropanol, and Milli-Q-water and then drying them with a stream of gaseous nitrogen. Next, silicone gaskets (CultureWell Reusable Gasket, Grace Bio-Labs) were placed on the clean glass coverslips to create defined wells. To convert optical reflection-interference contrast into a molecular mass, a known protein size marker (NativeMark Unstained Protein Standard, Invitrogen) was measured. For mass measurements, wells were filled with 18 µl measurement buffer (25 mM Tris-HCl, pH 7.5, 75 mM NaCl) to facilitate focusing the microscope onto the glass plate surface. Next, respective proteins/protein complexes were added into the well, and sample binding to the glass coverslip surface was monitored for 1 min using the AcquireMP software (Refeyn Ltd). Data analysis was carried out using DiscoverMP software (Refeyn Ltd).

## GLOE-seq

The protocol was modified based on the method described by Petrosino et al.[55]. All the oligonucleotides listed below were adapted from Ref. [55]. The nuclease reactions were performed with the indicated proteins and the scDNA (5.5 kbp) substrate as described above for the nicking assays in 15 µl volume. The reactions were stopped by incubation at 95 °C for 10 min, followed by cooling on ice for 5 min. Next, a ligation step was performed to capture the free 3′-OH termini. The reaction mixture included 14.85 nM DNA, 6.5 µl of 10x T4 DNA ligase buffer (Thermo Fisher Scientific), 1.4 µl of 5 µM stock proximal adapter (Integrated DNA Technologies), corresponding to 7.5-fold excess, 19.5 µl of 50% PEG 8000 (Thermo Fisher Scientific), 3 µl of T4 DNA ligase (Thermo Fisher Scientific), and water to reach a final volume of 65 µl. The thermal cycle was set up with the following conditions: annealing at 25 °C for 1 h and ligation at 22 °C for 2 h. The ligated product was purified using AMPure beads (Beckman Coulter) and eluted in 103 µl of water. The purified product was then sonicated using a Bioruptor Pico (20 cycles of 30 s on/30 s off). The sonicated product was purified twice with AMPure beads and once with streptavidin MyOne C1 Dynabeads (Thermo Fisher Scientific). The beads were washed once for 5 min with 1x SSC buffer (150 mM NaCl, 15 mM sodium citrate, pH 7.0), followed by wash with 20 mM NaOH for 10 min on a rotating wheel. Finally, DNA was eluted in 16 µl of water. For the synthesis of second-strand, the following reaction was set up in a PCR tube: 14.85 µl of eluate, 1.5 µl of oligonucleotide HU3790 (Integrated DNA Technologies), 0.3 unit of Phusion polymerase, 6 µl of 5x HF buffer (New England Biolabs; 200 mM Tris-HCl pH 7.4, 100 mM KCl, 100 mM $(NH_4)_2SO_4$, 20 mM $MgCl_2$, 1% Triton X-100), 5 µl of water, and 0.6 µl of 10 µM dNTP mix making a total volume of 30 µl. The reaction was incubated in a thermal cycler using the following program: denaturation at 95 °C for 2 min, annealing at 60 °C for 30 s, and extension at 72 °C for 2 min. The amplified dsDNA was purified using AMPure beads (Beckman Coulter). A subsequent polishing reaction performed with the NEBNext Ultra II DNA Library Prep Kit (New England Biolabs) with the following PCR conditions: 98 °C for 30 min; 98 °C for 10 min; 65 °C for 1 min 15 s (repeated 12 times), with a final step of 65 °C for 5 min before cooling. The resulting products were ligated with the distal adapter (Integrated DNA Technologies) using the reagents from NEBNext Ultra II DNA Library Prep Kit (New England Biolabs) for 20 min at 20 °C, followed by purification using AMPure beads and eluted in 20 µl of water. Finally, the DNA library was amplified

according to NEBNext Ultra II DNA Library Prep Kit (New England Biolabs) and sequenced using Nextseq 2000 using single-end 120 nt reads with dual 8-base indices.

The sequences obtained were aligned to the corresponding reference plasmid using a custom Python script (https://github.com/Sam18-hub/CutfinderOri). To accommodate the plasmid's circular nature, the reference sequence was duplicated and concatenated three times. A custom Python script (https://github.com/Sam18-hub/CutfinderOri) was used to determine the strand-specific cleavage events, where reads aligned to the top strand were used to infer cuts occurred on the bottom strand, and vice versa. The data resulting from the analysis, which contains the sequence and the mapped reads, were plotted in a polar plot.

## Structural modeling

Sequences of *H. sapiens MLH1* (P40692), *MLH3* (Q9UHC1), *EXO1* (Q9UQ84) and *MSH4* (O15457) were retrieved from the UniProt database[56] and were used as input of mmseqs2 homology search program[57] to generate a multiple sequence alignment (MSA) against the UniRef30 clustered database[58]. Homologs sharing less than 25% sequence identity with their respective query and less than 50% of coverage of the aligned region were not kept. In every MSA, in case several homologs belonged to the same species, only the one sharing the highest sequence identity to the query was kept. Full-length sequences of the orthologs were retrieved and re-aligned with mafft[59]. The size of the individual MSAs was then restricted following the delimitations indicated in Supplementary Table 2 to enhance the sensitivity of AlphaFold2 detection for interactions involving intrinsically disordered regions[49]. The delimited MSAs were concatenated so that when homologs of different subunits belonged to the same species, they were aligned in a paired manner, otherwise they were left unpaired[60]. The number of paired and unpaired sequences in the final concatenated MSAs is reported in Supplementary Table 2. Concatenated MSAs were used as inputs to generate 25 structural models (5 different seeds) for each of the conditions using a local version of the ColabFold v1.3 interface[61] running 3 iterations of the Alphafold2 v2.2.0 algorithm[43] trained on the multimer dataset[62]. Four scores were provided by AlphaFold2 to rate the quality of the models, the pLDDT, the pTMscore, the ipTMscore and the model confidence score (weighted combination of pTM and ipTM scores with a 20:80 ratio). The scores obtained for all the generated models are reported in Supplementary Table 3 for the model ranked first according to the confidence score. The models ranked first for each complex were relaxed using OpenMM engine and AMBER force field rosetta relax protocols to remove steric clashes[63] under constraints (std. dev. of 2 Å for the interatomic distances) and were deposited in the ModelArchive database (https://modelarchive.org/) (Supplementary Table 3). The Molecular graphics and analyses were performed with UCSF ChimeraX[64]. The conservation index shown in Supplementary Fig. 6d was calculated using the Rate4Site program[65] on the aligned sequences of each MutS family homolog. The alignment was built by selecting the first 120 orthologs aligned with the human protein after removing redundant sequences with more than 75% identity.

## Reporting summary

Further information on research design is available in the Nature Portfolio Reporting Summary linked to this article.

## Data availability

The structural models are available in ModelArchive (modelarchive.org) with the accession codes ma-m1f7g [https://www.modelarchive.org/doi/10.5452/ma-m1f7g] and ma-v510n [https://www.modelarchive.org/doi/10.5452/ma-v510n] for the MLH1-MLH3-EXO1 and MSH4-EXO1 complex, respectively. Movies underlying mass photometry analysis are uploaded to Dryad at https://doi.org/10.5061/

dryad.6m905qgbn. The FASTQ files generated from the sequencing of GLOE-seq libraries have been deposited in the European Nucleotide Archive (ENA) at EMBL-EBI (https://www.ebi.ac.uk/ena/browser/home) under the study accession number PRJEB88381. The raw read accession numbers are ERR14838039 and ERR14838040. Protein expression constructs are available on request from the corresponding authors. Source data are provided with this paper.

## Code availability

The codes used for data processing and analysis of GLOE-seq are available in the GitHub repository at https://github.com/Sam18-hub/CutfinderOri.

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

## Acknowledgements

We thank members of the Cejka laboratory for critical comments on the manuscript. We thank Josef Jiricny, Eric Alani and Ulrich Hübscher for protein expression constructs. Research in the Cejka laboratory is funded by the Swiss National Science Foundation (SNSF) (Grants 310030_207588 and 310030_205199) and the European Research Council (ERC) (Grant 101018257). R.G. was funded by Agence Nationale de la Recherche (ANR-21-CE44-0009) with granted access to the HPC resources of IDRIS under the allocation AD010314343R1, made by GENCI, and to the BIOI2 platform resources at the I2BC.

## Author contributions

M.R. and A.S. designed, performed and analyzed majority of the experiments. R.G. performed all structural modeling. I.S. purified MutLγ and performed GLOE-seq with the assistance of Ar.C. and A.R. J.S. and An.C. assisted with biophysical experiments that were not included in the paper. M.R., A.S., R.G. and P.C. wrote the manuscript, and P.C. supervised the study.

## Competing interests

The authors declare no competing interests.
