## [Transparent Peer Review file · Nature Communications]

EXO1 promotes the meiotic MLH1-MLH3 endonuclease through conserved interactions with MLH1, MSH4 and DNA

Corresponding Author: Professor Petr Cejka

Version 0:

Reviewer comments:

Reviewer #1

(Remarks to the Author)

In this manuscript, Sanchez and colleagues investigated the mechanism whereby EXO1 stimulates the DNA cleavage activity of the meiotic MLH1-MLH3 endonuclease. EXO1 directly interacts with MLH1, but the stimulatory activity of EXO1 on DNA cleavage depends on MSH4-MSH5. Using AlphaFold modeling, the authors identified key residues within EXO1 and MSH4 responsible for this stimulatory activity. In addition, the authors show that EXO1-dependent stimulation of DNA cleavage also requires DNA binding by EXO1.

This is a really lovely paper. The results are exciting and the data are very robust. The biochemical work is extensive, manuscript is clearly written, the figures are neat, all the data are quantified and stats appropriately described. Aside for two typos (L67 : Contrast, L882 : Thank), I couldn't find any issue with the paper at all.

Therefore, in my opinion, the paper can be published essentially as it is. I have a couple of minor questions/suggestions, but addressing these should in no way be regarded as essential for publication.

- The authors tested whether a synthetic peptide corresponding to residues 353-390 of EXO1 is sufficient to stimulate the cleavage activity of MLH1-MLH3 and found that it is not the case. Would the peptide prevent EXO1-dependent stimulation in a competition experiment?

- I know it is obvious from the sequence alignments, but since the title emphasizes the conservation of the interaction, I think it would be nice to show a selection of AlphaFold models from different species of the EXO1 – MSH4, EXO1-MLH1 and EXO1-DNA interactions in the supplement.

- Figure 2A and 5A, I suggest to label Exo1 on the figure to make it more self-explanatory.

Reviewer #2

(Remarks to the Author)

In this manuscript, Sanchez et. al characterize how EXO1 stimulates MutL γ endonuclease activity to function in meiotic recombination, an activity that was previously shown to be dependent on MutS γ and RFC-PCNA. The authors developed conditions that allowed them to examine the effect of EXO1 on stimulated/activated MutL γ nicking activity in the absence of RFC-PCNA; this activity remained dependent on MutS γ , indicating some specificity. The authors identified key residues within EXO1 that are responsible for direct interactions with MutL γ . Notably, disrupting this interaction did not eliminate EXO1's stimulatory activity, leading the authors to propose the EXO1 interacts directly with MutS γ . They demonstrated this EXO1-MutS γ interaction, identified series of residues within Msh4 and EXO1 that support this interaction. In contrast to the interaction the MutL γ , mutations that disrupted the MutS γ -EXO1 interaction abrogated stimulation of MutL γ endonuclease activity. Mutations of these residues within EXO1 did not compromise the intrinsic nicking activity of EXO1, leading the authors to conclude that EXO1's stimulation of MutL γ activity is separate from its intrinsic nuclease activity. Lastly, they identified EXO1 residues involved in DNA binding, and confirmed they were necessary for MutL γ stimulation. Overall, this work is a careful analysis of EXO1 stimulation of MutL γ endonuclease activity and reveals EXO1-MuS γ and EXO1-DNA interaction as essential components of this process, which is critical for meiotic crossover control. Overall the results have important implications for our understanding of meiotic recombination. We have some comments to strengthen and clarify

the arguments made in the manuscript.

1. In Figure 2, the analysis of the direct physical interaction between EXO1 and MutLy is complicated by the fairly high background binding of EXO1 to the beads/column. Further, the authors indicate that the interaction is disrupted in the D173A, I403E, MIP mutant, although there does appear to be significant pull-down of EXO1, albeit reduced relative to other constructs. Therefore, the conclusions from this section warrant some quantification, recognizing that a pull-down has limitations in this regard. Nonetheless, it is important to quantify the levels of EXO1 +/- MutLy, perhaps including normalization via the Mlh1 signal, for a more accurate assessment of the extent of the interactions.

With the EXO1 triple mutant, which, as noted above, the authors conclude is disrupted for its interaction with MutLy, the authors note only a slight reduction in stimulation of MutLy. They conclude from this that EXO1's interaction with MutLy is not necessary for stimulation. An alternative possibility is that the extent of stimulation is correlated to the extent of stimulation, pending further analysis of the pull-down data. Therefore, the conclusions from this figure should be tempered somewhat, with mention that some interaction remained, and this may be responsible for the resulting stimulation. Additionally, it would strengthen their argument to examine mutants that exhibit this complete loss of EXO1 and MutLy interaction to measure the impact on MutLy stimulation.

Finally, a second method for characterizing and quantifying the EXO1-MutLy interaction would strengthen the conclusions here. For example, the authors use mass photometry to interrogate the EXO1-MutLy interaction.

2. We recognize that this is a somewhat artificial system, but assessing the role of DNA in these interactions seems relevant, particularly since that is the read-out for the MutLy-MutSy-EXO1 ensemble (+/-RFC-PCNA). DNA interactions could interfere or enhance protein-protein interactions. Have the authors done any interaction assays in the presence of DNA?

3. Quantification and statistical analyses of pull-downs and in vitro assays are missing throughout the manuscript. These analyses would strengthen the authors' arguments in several places.

4. It would be useful to test the effects of the EXO1 triple mutant on interactions with MutSy and of the EXO1 EE or EEE mutants on interactions with MutLy. These residues are relatively close to the I403 position, at least, and there could be some reciprocal influence on protein-protein interactions that could, in turn, affect stimulation of MutLy endonuclease activity.

5. In the discussion, the authors note a proposed alternative role for EXO1 in meiotic recombination than was identified in yeast, where it functions to prevent ligation of nicks to ensure their maturation. Some rewording to highlight further that both functions may be performed by EXO1 and their work does not exclude the possibility of additional functions of EXO1 in humans should be considered. The model for the role of EXO1 in enhancing the MutLy ensemble is a little unclear. Is it proposed to stabilize interactions to stimulate the endonuclease activity? Does it facilitate a potential oligomerization of the MutLy complex or alter the ATPase activity of either MutSy or MutLy? A little more discussion would clarify how the work fits into the broader field.

Minor comments:

Please indicate the residue changes for the MIP mutants in the text (p. 5) for clarity.

Does the MSH4 residues involved in an interaction with EXO1 align with conserved residues in other MutS homologues?

It is unclear what is meant in the second half of the sentence on lines 255-257.

Reviewer #3

(Remarks to the Author)
NCOMMS-24-72027

The paper by Sanchez et al. shows the biochemical characterization of the role of human Exo1 nuclease in the resolution of double-Holliday junction by human Mlh1-Mlh3 (MutLgamma) nucleases through the interaction with human Msh4-Msh5 (MutSgamma) as well as DNAs. The authors nicely showed that Exo1 and MutSg work together to stimulate MutLgamma's nicking activity. Moreover, the authors identified key residues and regions on Exo1 and Msh4 for the stimulative activity. They conclude that the interaction between Exo1 and MutSgamma as well as the DNA-binding activity of Exo1 are critical for the nicking activity. Combined with AlphaFold2 prediction, the authors provided very much convincing in vitro results and the text is well-written with a nice figure presentation. The lack of in vivo experiments is a big concern although yeast Exo1 does not have the conserved activity on MutSgamma as human Exo1. The weakness is that the authors did not provide molecular insight on how Exo1 stimulates MutLgamma nicking activity through the interaction with MutSgamma. Given the DNA binding activities (N-terminal and C-terminal regions) of Exo1 is critical for the stimulative activity, the authors need more careful quantification of Exo1's DNA binding (see major point #2).

Major points:

1. Given the interaction of Exo1 with MutSgamma, it is very interesting to check whether the 5'-3' exonuclease activity of

Exo1 is modulated by the MutSgamma. Moreover, although the Exo1 variants shown in Figure 3 bind to dsDNA, it is very important to the exonuclease activity of these variants of Exo1.

2. In the same line, MutSgamma binds to a branched DNA substrate such Holliday junction (HJ). It is worthwhile if the authors examined the effect of Exo1 on the binding of the HJ such as the stabilization of the DNA-protein complex (by EMSA etc).
3. DNA binding assay, Figure 3I, 7D, and S4A: To clarify the binding activities of Exo1 variant, rather than the EMSA, the author can use the fluorescence anisotropy to measure the K_d values. This is very important since the authors discuss a possible role of the Exo1 DNA binding in the stimulation of the MutLgamma. And also the authors can use the nuclease-deficient versions of Exo1 (Figure 4) in the DNA binding assay.
4. The yeast MutSgamma binds to yeast Exo1 or not? If the authors would show this, the in vivo experiments would be possible.
5. Although not essential, the nicking assay used in this study does not provide any DNA sequence information on the product. Thus, in each reaction, the nature of ends of nicking products may be different. The authors may be able to analyze the sequence by the Linker-mediated ligation (and PCR amplification) followed by the NGS sequencing. Or just, the authors check whether the nick is sealed or not by the Ligase or not.

Minor points:

1. Line 163, Exo1-MIP: Please show what kind of amino acid substitution or deletion for this mutant Exo1.
2. Line 176-177, the human EXO1 (D173A I403E MIP) variant is impaired in its physical interaction with MutLgamma, it is still able to notably stimulate the MutSgamma-MutLgamma nuclease in a species-specific manner: Given Figure 2E (the mutant Exo1 binds well with MutLgamma), this statement is too strong. It is better to rephrase.
3. Figure 5L: Do the WT peptide interfere with the nicking activity of MutLgamma by MutSgamma?
4. Figure 7E: What happens when Mn²⁺ without PCNA-RFA is used?

Reviewer #4

(Remarks to the Author)

Version 1:

Reviewer comments:

Reviewer #1

(Remarks to the Author)

The authors have appropriately addressed all the comments raised by the reviewers. Therefore, I wholeheartedly support the publication of the manuscript in its current form and congratulate the authors for their nice work.

Reviewer #2

(Remarks to the Author)

We are satisfied with the revised version of this manuscript. The authors have addressed our concerns and comments, which has improved the rigour of the work. We appreciate their efforts.

Reviewer #3

(Remarks to the Author)

The authors addressed the previous points properly except for the binding assay using double-Holliday junction and also added new data such as GLOE-seq. There are some minor comments.

1. Line 143, GLOE-seq: Please explain what the GLOE-seq is for the readers in the main text and Figure legend.
2. Line 299, to promote the MutSg-MutLg and RFC-PCNA “ensemble”: The “ensemble” is a bit obscure here. Rather, “to promote the nicking activity by the MutSg-MutLg and RFC-PCNA” is a more appropriate word here.
3. Supplementary Figure 3: Please use the species name rather than the protein accession numbers.

Reviewer #4

(Remarks to the Author)

Manuscript title: EXO1 promotes the meiotic MLH1-MLH3 endonuclease through conserved interactions with MLH1, MSH4 and DNA

We would like to thank the reviewers for their valuable comments and suggestions. All conclusions remain unchanged, but we were able to strengthen the manuscript in several points including:

- We provide quantifications of protein interaction assays where wild-type and mutant variants were compared, and include statistics for comparisons between relevant samples in bar graphs
- We provide an orthogonal validation for EXO1-DNA binding to complement EMSA data
- We used GLOE-seq to monitor DNA incisions by MutL γ with or without MutS γ , EXO1, RFC and PCNA.
- We provide computational analysis of the conservation of the modelled interaction sites
- We hypothesize that the dsDNA binding capacity of EXO1 might keep the MutS γ -MutL γ tethered dsDNA downstream of recognition of HJs, explaining the structural role of EXO1

Below we provide the detailed responses to the individual comments.

Reviewer #1 (Remarks to the Author):

In this manuscript, Sanchez and colleagues investigated the mechanism whereby EXO1 stimulates the DNA cleavage activity of the meiotic MLH1-MLH3 endonuclease. EXO1 directly interacts with MLH1, but the stimulatory activity of EXO1 on DNA cleavage depends on MSH4-MSH5. Using AlphaFold modeling, the authors identified key residues within EXO1 and MSH4 responsible for this stimulatory activity. In addition, the authors show that EXO1-dependent stimulation of DNA cleavage also requires DNA binding by EXO1.

This is a really lovely paper. The results are exciting and the data are very robust. The biochemical work is extensive, manuscript is clearly written, the figures are neat, all the data are quantified and stats appropriately described. Aside for two typos (L67 : Contrast, L882 : Thank), I couldn't find any issue with the paper at all.

Therefore, in my opinion, the paper can be published essentially as it is. I have a couple of minor questions/suggestions, but addressing these should in no way be regarded as essential for publication.

- The authors tested whether a synthetic peptide corresponding to residues 353-390 of EXO1 is sufficient to stimulate the cleavage activity of MLH1-MLH3 and found that it is not the case. Would the peptide prevent EXO1-dependent stimulation in a competition experiment?

REPLY: We thank the reviewer for the positive comments on our work. We performed additional experiments to assess whether the synthetic peptide could prevent EXO1-dependent stimulation of MLH1-MLH3. We did not observe any effect when using the 353-390 peptide in a competition experiment (Fig. R1). As this result has multiple potential explanations, we decided not to include it in the manuscript.

Figure R1. Nicking assays with indicated proteins in the presence or absence of a peptide corresponding to EXO1 residues 353 to 390. N=3, error bars, SEM.

- I know it is obvious from the sequence alignments, but since the title emphasizes the conservation of the interaction, I think it would be nice to show a selection of AlphaFold models from different species of the EXO1 – MSH4, EXO1-MLH1 and EXO1-DNA interactions in the supplement.

REPLY: We performed a comparative structural analysis of the predicted interface between MSH4 and EXO1 in two additional vertebrate species: *Xenopus laevis* and *Danio rerio*. The residues of the SIW motif in EXO1 are invariant in *Homo sapiens*, *Xenopus laevis* and *Danio rerio*, while the human Y375 is not conserved in the other species. On the MSH4 side, most of the observed substitutions are conservative and the major hydrophobic residues are conserved. These data are now included as new Fig. S6c.

- Figure 2A and 5A, I suggest to label Exo1 on the figure to make it more self-explanatory.

REPLY: We have added labels for EXO1 alignments in both the figures.

Reviewer #2 (Remarks to the Author):

In this manuscript, Sanchez et. al characterize how EXO1 stimulates MutLy endonuclease activity to function in meiotic recombination, an activity that was previously shown to be dependent on MutSy and RFC-PCNA. The authors developed conditions that allowed them to examine the effect of EXO1 on stimulated/activated MutLy nicking activity in the absence of RFC-PCNA; this activity remained dependent on MutSy, indicating some specificity. The authors identified key residues within EXO1 that are responsible for direct interactions with MutLy. Notably, disrupting this interaction did not eliminate EXO1's stimulatory activity, leading the authors to propose the EXO1 interacts directly with MutSy. They demonstrated this EXO1-MutSy interaction, identified series of residues within Msh4 and EXO1 that support this interaction. In contrast to the interaction the MutLy, mutations that disrupted the MutSy-EXO1 interaction abrogated stimulation of MutLy endonuclease activity. Mutations of these residues within EXO1 did not compromise the intrinsic nicking activity of EXO1, leading the authors to conclude that EXO1's stimulation of MutLy activity is separate from its intrinsic nuclease activity. Lastly, they identified EXO1 residues involved in DNA binding, and confirmed they were necessary for MutLy stimulation. Overall, this work is a careful analysis of EXO1 stimulation of MutLy endonuclease activity and reveals EXO1-MuSy and EXO1-DNA interaction as essential components of this process, which is critical for meiotic crossover control. Overall the results have important implications for our understanding of meiotic recombination. We have some comments to strengthen and clarify the arguments made in the manuscript.

1. In Figure 2, the analysis of the direct physical interaction between EXO1 and MutLy is complicated by the fairly high background binding of EXO1 to the beads/column. Further, the authors indicate that the interaction is disrupted in the D173A, I403E, MIP mutant, although there does appear to be significant pull-down of EXO1, albeit reduced relative to other constructs. Therefore, the conclusions from this section warrant some quantification, recognizing that a pull-down has limitations in this regard. Nonetheless, it is important to quantify the levels of EXO1 +/- MutLy, perhaps including normalization via the Mlh1 signal, for a more accurate assessment of the extent of the interactions.

With the EXO1 triple mutant, which, as noted above, the authors conclude is disrupted for its interaction with MutLy, the authors note only a slight reduction in stimulation of MutLy. They conclude from this that EXO1's interaction with MutLy is not necessary for stimulation. An alternative possibility is that the extent of stimulation is correlated to the extent of stimulation, pending further analysis of the pull-down data. Therefore, the conclusions from this figure should be tempered somewhat, with mention that some interaction remained, and this may be responsible for the resulting stimulation. Additionally, it would strengthen their argument to examine mutants that exhibit this complete loss of EXO1 and MutLy interaction to measure the impact on MutLy stimulation.

REPLY: As suggested, we performed additional repeats, quantified and statistically analysed the interaction of the EXO1 variants with MutLy (Fig. 2d). The background binding of EXO1 to MLH1 antibody coated beads was subtracted and the level of EXO1 variants was normalized to the corresponding MLH1 signal. Additionally, we have also added statistical analyses to the nicking assays in Fig 2e. The conclusion is that only the impairment of the physical interaction with the triple mutant is statistically significant (Fig. 2d), while the effect in nicking assays is not significant (Fig. 2e). Nevertheless, we agree with the reviewer's comment, and we note in the relevant section of the results that the nuclease activity may be at least in part dependent on the residual interaction.

Finally, a second method for characterizing and quantifying the EXO1-MutLy interaction would strengthen the conclusions here. For example, the authors use mass photometry to interrogate the EXO1-MutLy interaction.

REPLY: We agree with the reviewer and we spent a significant amount of time trying to address this point. We do not have sufficient amounts and concentrations of EXO1 and MutLy for biophysical experiments such as SPR or BLI (see our response to Reviewer # 3). In mass photometer, EXO1 and MutLy are largely monodisperse (as shown in the manuscript), but MutLy unfortunately tends to partially fall apart into the individual subunits, and shows some oligomerization. Because the interaction between EXO1 and MutLy is not very strong (strongly substoichiometric), we were not able to reliably detect the interaction peak of EXO1-MutLy considering the strong background coming from the MutLy sample.

2. We recognize that this is a somewhat artificial system, but assessing the role of DNA in these interactions seems relevant, particularly since that is the read-out for the MutLy-MutLy-EXO1 ensemble (+/-RFC-PCNA). DNA interactions could interfere or enhance protein-protein interactions. Have the authors done any interaction assays in the presence of DNA?

REPLY: The reviewer is correct but there are technical difficulties. Given that each protein can independently bind DNA, including DNA in the protein interaction assays may not be informative, because the proteins might simultaneously bind the same DNA molecule rather than to interact with each other. We have also attempted to address this issue using electrophoretic mobility shift assays (see response to Reviewer #3, point 2).

3. Quantification and statistical analyses of pull-downs and in vitro assays are missing throughout the manuscript. These analyses would strengthen the authors' arguments in several places.

REPLY: We agree with the reviewer's comment. We have added quantifications of all protein pull-down assays where wild type and mutant protein variants are compared. Additionally, we have also added statistics to a number of relevant panels throughout the manuscript.

4. It would be useful to test the effects of the EXO1 triple mutant on interactions with MutSy and of the EXO1 EE or EEE mutants on interactions with MutLy. These residues are relatively close to the I403 position, at least, and there could be some reciprocal influence on protein-protein interactions that could, in turn, affect stimulation of MutLy endonuclease activity.

REPLY: As suggested, we performed pull-down of EXO1 (D173A MIP I403E) with MutSy immobilizing the EXO1 variant via FLAG tag, and pull-down of EXO1 EE (W371E Y375E) and EXO1 EEE (S369E W371E Y375E) variants with MutLy. As expected, the interactions were not reduced by the tested mutations (Fig. R2).

Figure R2: Left, protein interaction assays of MutLy and EXO1 variants WT, EE (W371E Y375E) and EEE (S369E W371E W375E). MutLy (bait) was immobilized using an anti-MLH1 antibody, and EXO1 variants (preys) were subsequently added. Top, a schematic of the assay. Bottom, representative experiments analyzed by Western blotting. Right, protein interaction assays of MutSy and EXO1 (D173A MIP I403E). The EXO1 variant (bait) was immobilized using an anti-FLAG antibody, and MutSy (prey) were subsequently added. F506A and F507A mutations within the MIP motif are referred to as the MIP mutations. Top, a schematic of the assay. Bottom, representative experiments analyzed by Western blotting.

5. In the discussion, the authors note a proposed alternative role for EXO1 in meiotic recombination than was identified in yeast, where it functions to prevent ligation of nicks to ensure their maturation. Some rewording to highlight further that both functions may be performed by EXO1 and their work does not exclude the possibility of additional functions of EXO1 in humans should be considered. The model for the role of EXO1 in enhancing the MutLy ensemble is a little unclear. Is it proposed to stabilize interactions to stimulate the endonuclease activity? Does it facilitate a potential oligomerization of the MutLy complex or alter the ATPase activity of either MutSy or MutLy? A little more discussion would clarify how the work fits into the broader field.

REPLY: We performed an ATPase assay but unfortunately the ATPase activity of MutSy and MutLy was too low to be quantified (Fig. R3). However, based on new electrophoretic mobility shift assays with HJs and dsDNA (Supplementary Fig. 7d,e) and published data (PMID: 32814904, 15304223, 24443562, 29625041), we suggest that EXO1 might help keep the MutSy-MutLy complex bound to dsDNA downstream of the recognition of a HJ or a related structure. Please see also our response to Reviewer #3, point 2 and the Discussion text for more details.

Figure R3: ATP hydrolysis assay with indicated protein (in the presence of scDNA, 1 μM, in nucleotides). Sgs1 was used as a positive control.

Minor comments:

Please indicate the residue changes for the MIP mutants in the text (p. 5) for clarity.

REPLY: The residue changes for the MIP mutation (F506A and F507A) are now added in the text.

Does the MSH4 residues involved in an interaction with EXO1 align with conserved residues in other MutS homologues?

REPLY: We did a comparative analysis of the domains in MutS family proteins homologous to MSH4 in humans (new Fig. S6d). Interestingly, the region of MSH4 involved in binding exhibits a distinct conserved pocket primarily composed of hydrophobic residues. This contrasts with other homologs of the MutS family, which also have conserved positions in this region but include at least two charged residues. We have modified the main text accordingly.

It is unclear what is meant in the second half of the sentence on lines 255-257.

REPLY: We have rephrased the text.

Reviewer #3 (Remarks to the Author):

NCOMMS-24-72027

The paper by Sanchez et al. shows the biochemical characterization of the role of human Exo1 nuclease in the resolution of double-Holliday junction by human Mlh1-Mlh3 (MutLgamma) nucleases through the interaction with human Msh4-Msh5 (MutSgamma) as well as DNAs. The authors nicely showed that Exo1 and MutSg work together to stimulate MutLgamma’s nicking activity. Moreover, the authors identified key residues and regions on Exo1 and Msh4 for the stimulative activity. They conclude that the interaction between Exo1 and MutSgamma as well as the DNA-binding activity of

Exo1 are critical for the nicking activity. Combined with AlphaFold2 prediction, the authors provided very much convincing in vitro results and the text is well-written with a nice figure presentation. The lack of in vivo experiments is a big concern although yeast Exo1 does not have the conserved activity on MutSgamma as human Exo1. The weakness is that the authors did not provide molecular insight on how Exo1 stimulates MutLgamma nicking activity through the interaction with MutSgamma. Given the DNA binding activities (N-terminal and C-terminal regions) of Exo1 is critical for the stimulative activity, the authors need more careful quantification of Exo1's DNA binding (see major point #2).

Major points:

1. Given the interaction of Exo1 with MutSgamma, it is very interesting to check whether the 5'-3' exonuclease activity of Exo1 is modulated by the MutSgamma. Moreover, although the Exo1 variants shown in Figure 3 bind to dsDNA, it is very important to the exonuclease activity of these variants of Exo1.

REPLY: We observed that the exonuclease activity of EXO1 is slightly inhibited in the presence of either MutSy or MutLy. The same effect was observed with γ Exo1. These data suggest that DNA binding by MutSy or MutLy may non-specifically limit the accessibility of EXO1 to DNA. We believe that stimulation would have been unexpected, considering that the nuclease activity of Exo1 is not needed to resolution and crossover formation.

Fig R3: Nuclease assay using 3'-labeled 50 bp dsDNA with indicated proteins. The reaction products were analyzed by denaturing PAGE.

2. In the same line, MutSgamma binds to a branched DNA substrate such Holliday junction (HJ). It is worthwhile if the authors examined the effect of Exo1 on the binding of the HJ such as the stabilization of the DNA-protein complex (by EMSA etc).

REPLY: We thank the reviewer for this suggestion. According to the literature, MutSy and MutLy preferentially bind branched structures such as Holliday junctions, with MutLy having a higher affinity than MutSy (e.g. PMID: 32814904, 15304223). Accordingly, under our conditions, the preferential binding of MutLy to HJs over dsDNA was clearly apparent (new Supplementary Fig. 7e). In contrast to MutSy and MutLy, EXO1 clearly binds also dsDNA (new Supplementary Fig. 7d). While MutSy and MutLy are thought to be recruited to HJs (e.g. PMID: 32814904, 15304223, 24443562), but then are likely to cleave some distance away (29625041), one possibility is that EXO1 helps keep the MutSy-MutLy complex bound to dsDNA downstream of the recognition of a HJ or a related structure. We have added this speculative model in the Discussion, which is further supported by our data that dsDNA binding by EXO1 is necessary for EXO1's function in the stimulation of MutSy-MutLy, as well as partially by data from the Alani lab (37079643).

3. DNA binding assay, Figure 3I, 7D, and S4A: To clarify the binding activities of Exo1 variant, rather than the EMSA, the author can use the fluorescence anisotropy to measure the Kd values. This is very

important since the authors discuss a possible role of the Exo1 DNA binding in the stimulation of the MutLgamma. And also the authors can use the nuclease-deficient versions of Exo1 (Figure 4) in the DNA binding assay.

REPLY: We attempted to monitor DNA binding by microscale thermophoresis (Fig. R4). As seen in the panel, the available concentrations of our EXO1 variants are below what is necessary to saturate the curves and obtain reliable K_d values. We were similarly unsuccessful in SPR experiments - we immobilized the EXO1 variants and titrated DNA, yet also here we could not get reliable data (not shown). However, we managed to orthogonally validate the EMSA experiments in pulldown assays, where we immobilized dsDNA on streptavidin beads, and incubated with the EXO1 variants. In accord with EMSA, the KD KD mutations only marginally affected the DNA binding of full-length EXO1, while the same mutations largely abolished DNA binding of the short 1-390 EXO1 truncation construct (new Fig. 7e). Regarding the last point, in Fig. 3I, we monitored DNA binding by nuclease-deficient (D173A) EXO1 truncation variants, and the data are very consistent with the rest of the manuscript where nuclease-proficient EXO1 and mutants were used (with EDTA).

Fig R4: MST curves with different EXO1 variants. The dsDNA (0.5 nM) was Cy5-labelled and EXO1 variants were used at increasing concentrations. When higher DNA concentration was used, the curves started rising even later.

4. The yeast MutSgamma binds to yeast Exo1 or not? If the authors would show this, the in vivo experiments would be possible.

REPLY: We could barely detect the interaction of yeast MutSy and Exo1 (Fig. S3a). Furthermore, the interaction motif downstream of the nuclease domain is conserved in vertebrates but not in yeast (Fig. 2a), so the observed interplay cannot be tested using this model system.

5. Although not essential, the nicking assay used in this study does not provide any DNA sequence information on the product. Thus, in each reaction, the nature of ends of nicking products may be different. The authors may be able to analyze the sequence by the Linker-mediated ligation (and PCR amplification) followed by the NGS sequencing. Or just, the authors check whether the nick is sealed or not by the Ligase or not.

REPLY: Thank you for the comment. We have used GLOE-seq to detect the positions of the DNA incision points by MutLy without or with co-factors (new Fig. 1a). We show that the incision sites are distributed all over the plasmid, but clearly some sites are cleaved better than others. Additionally, as suggested, we treated the nicked products with T4 ligase, and we observed that a fraction of the nicked products can be ligated (Fig. R5).

Figure R5. Nicking assays with indicated proteins either in the presence of 3 mM Mn²⁺ (MutLy only), or in the presence of 0.6 mM Mg²⁺ (complete ensemble) (lanes 1-3). The reaction products were purified and treated with T4 ligase (lanes 4-6). Please note that the left 3 lanes are included in the manuscript (Fig. 1a).

Minor points:

1. Line 163, Exo1-MIP: Please show what kind of amino acid substitution or deletion for this mutant Exo1.

REPLY: We have added the amino acid substitution for MIP mutation (F506A and F507A) in the text.

2. Line 176-177, the human EXO1 (D173A I403E MIP) variant is impaired in its physical interaction with MutLgamma, it is still able to notably stimulate the MutSgamma-MutLgamma nuclease in a species-specific manner: Given Figure 2E (the mutant Exo1 binds well with MutLgamma), this statement is too strong. It is better to rephrase.

REPLY: Please see our response to comment 1 of Reviewer #2. In updated Fig. 2d, we provide quantifications of protein interaction experiments, and in updated Fig. 2e we include statistics for the nuclease assays. The manuscript text has been adjusted accordingly.

3. Figure 5L: Do the WT peptide interfere with the nicking activity of MutLgamma by MutSgamma?

REPLY: Please see our response to comment 1 of Reviewer # 1.

4. Figure 7E: What happens when Mn²⁺ without PCNA-RFA is used?

REPLY: We decided to show the data with magnesium only as it is the more physiologically relevant metal cofactor and the reaction includes the complete ensemble. Nonetheless, we have observed that the effect of the respective EXO1 variants on the stimulation of MutLy-MutSy endonuclease activity is the same in manganese (Fig. R6).

Figure R6. Quantification of the nicking activity of the MutLy-MutSy complex and its stimulation by EXO1, EXO1 (D173A), EXO1 (D78A D173A) and EXO1 (K185D K237D) at 0.6 mM Mn²⁺. Averages shown, n = 6; error bars, SEM.

Reviewer #4 (Remarks to the Author):
